# Artificial intelligence-rationalized balanced PPARα/γ dual agonism resets dysregulated macrophage processes in inflammatory bowel disease

Gajanan D. Katkar[1], Ibrahim M. Sayed[2,3], Mahitha Shree Anandachar[2], Vanessa Castillo[1], Eleadah Vidales[1], Daniel Toobian[1], Fatima Usmani[2], Joseph R. Sawires[4], Geoffray Leriche[4], Jerry Yang[4], William J. Sandborn [5✉], Soumita Das [2✉], Debashis Sahoo [6,7,8✉] & Pradipta Ghosh [1,5,8,9✉]

A computational platform, Boolean network explorer (BoNE), has recently been developed to infuse AI-enhanced precision into drug discovery; it enables invariant Boolean Implication Networks of disease maps for prioritizing high-value targets. Here we used BoNE to query an Inflammatory Bowel Disease (IBD)-map and prioritize a therapeutic strategy that involves dual agonism of two nuclear receptors, PPARα/γ. Balanced agonism of PPARα/γ was predicted to modulate macrophage processes, ameliorate colitis, 'reset' the gene expression network from disease to health. Predictions were validated using a balanced and potent PPARα/γ-dual-agonist (PAR5359) in *Citrobacter rodentium*- and DSS-induced murine colitis models. Using inhibitors and agonists, we show that balanced-dual agonism promotes bacterial clearance efficiently than individual agonists, both in vivo and in vitro. PPARα is required and sufficient to induce the pro-inflammatory cytokines and cellular ROS, which are essential for bacterial clearance and immunity, whereas PPARγ-agonism blunts these responses, delays microbial clearance; balanced dual agonism achieved controlled inflammation while protecting the gut barrier and 'reversal' of the transcriptomic network. Furthermore, dual agonism reversed the defective bacterial clearance observed in PBMCs derived from IBD patients. These findings not only deliver a macrophage modulator for use as barrier-protective therapy in IBD, but also highlight the potential of BoNE to rationalize combination therapy.

[1] Department of Cellular and Molecular Medicine, University of California San Diego, San Diego, USA. [2] Department of Pathology, University of California San Diego, San Diego, USA. [3] Department of Medical Microbiology and Immunology, Faculty of Medicine, Assiut University, Assiut, Egypt. [4] Department of Chemistry and Biochemistry, University of California San Diego, San Diego, USA. [5] Department of Medicine, University of California San Diego, San Diego, USA. [6] Department of Computer Science and Engineering, Jacob's School of Engineering, University of California San Diego, San Diego, USA. [7] Department of Pediatrics, University of California San Diego, San Diego, USA. [8] Rebecca and John Moore Comprehensive Cancer Center, University of California San Diego, San Diego, USA. [9] Veterans Affairs Medical Center, La Jolla, San Diego, USA. ✉email: wsandborn@health.ucsd.edu; sodas@ucsd.edu; dsahoo@ucsd.edu; prghosh@ucsd.edu

nflammatory bowel disease (IBD) is an autoimmune disorder of the gut in which diverse components including microbes, genetics, environment, and immune cells interact in elusive ways to culminate in overt diseases[1–3]. It is also heterogeneous with complex sub-disease phenotypes (i.e., strictures, fistula, abscesses, and colitis-associated cancers)[4,5]. Currently, patients are offered anti-inflammatory agents that have a ~30–40% response rate, and 40% of responders become refractory to treatment within one year[6,7]. Little is known to fundamentally tackle the most widely recognized indicator/predictor of disease relapse i.e., a compromised mucosal barrier. Homeostasis within this mucosal barrier is maintained by our innate immune system, and either too little or too much reactivity to invasive commensal or pathogenic bacteria, is associated with IBD[8]. Although defects in the resolution of intestinal inflammation have been attributed to altered monocyte–macrophage processes in IBD, macrophage modulators are yet to emerge as treatment modalities in IBD[8].

We recently developed and validated an AI-guided drug discovery pipeline that uses large transcriptomic datasets (of the human colon) to build a Boolean network of gene clusters[9] (Fig. 1; Step 0); this network differs from other computational methods (e.g., Bayesian and Differential Expression Analyses) because gene clusters here are interconnected by directed edges that represent Boolean implication relationships (BIRs) that invariably hold true in every dataset within the cohort. Once built, the network is queried using machine learning approaches to identify in an unbiased manner which clusters most effectively distinguish healthy from diseased samples and do so reproducibly across multiple other cohorts (906 human samples, 234 mouse samples). Gene-clusters that maintain the integrity of the mucosal barrier emerged as the genes that are invariably downregulated in IBD, whose pharmacologic augmentation/induction was predicted to 'reset' the network. These insights were exploited to prioritize one target, choose appropriate pre-clinical murine models for target validation, and design patient-derived organoid models (Fig. 1; Step 0)[9]. Treatment efficacy was confirmed in patient-derived organoids using multivariate analyses. This AI-assisted approach provided an epithelial barrier-protective agent in IBD and predicted Phase-III success with higher accuracy over traditional approaches[9].

Here we use the same AI-guided drug discovery pipeline to identify and validate a macrophage modulator that is predicted to restore mucosal barrier and homeostasis in IBD (Fig. 1; Steps 1). Using primary peritoneal macrophages and specific agonists and antagonists, we reveal the mechanism(s) of action that enable balanced agonists of this pair of nuclear receptors, PPARα/γ, to reverse some of the fundamental imbalances of the innate immune system in IBD, such that immunity can be achieved without overzealous inflammation (Fig. 1; Step 2). We demonstrate the accuracy and predictive power of this network-rationalized approach and reveal the efficacy of balanced-dual agonists of PPARα/γ in two pre-clinical murine models (Fig. 1; Step 3) and in patient-derived PBMCs (Fig. 1; Step 4).

## Results
### Development of a web-based platform for generating a 'target report card'. We first developed an interactive, user-friendly web-based platform that allows the querying of our Boolean network-based-IBD map with the goal of enabling researchers to pick high-value targets[9] (Fig. 1; Step 0; Supplementary Fig. 1). The platform generates a comprehensive automated report containing actionable information for target validation, a 'target report card', which contains predictions on five components (Fig. 2a): (i) Impact on the outcome of IBD in response to treatment, which shows how levels of expression of any proposed target gene(s)

relates to the likelihood of response to therapies across diverse cohorts; (ii) Therapeutic index, a computationally generated index using Boolean implication statistics which provides a likelihood score of indicate whether pharmacologic manipulation of the target gene(s) would lead to success in Phase III clinical trials; (iii) Appropriateness of preclinical mouse models, a component that indicates which murine models of colitis shows the most significant change in the target genes (and hence, likely to be best models to test the efficacy of any manipulation of that target); (iv) Gender bias, a component that indicates whether the gene is differentially expressed in IBD-afflicted men versus women; and (v) Target tissue/cell type specificity, which shows the likely cell type where the target is maximally expressed, and hence, the cell type of desirable pharmacologic action. Details of how therapeutic index is computed are outlined in Methods and in Supplementary Fig. 2; it is essentially a statistical score of how tightly any proposed target gene(s) associates with FDA-approved targets versus those that failed and serve as an indicator of the likelihood of success[9]. Similarly, details of how cell type of action is computer are outlined in Methods and in Supplementary Fig. 3.

### PPARα/γ dual agonists are predicted to be effective barrier-protective agents in IBD. Previous work had identified a little over 900 genes in 3 clusters (Clusters #1-2-3 within the IBD map; Fig. 1: Step 0; Supplementary Fig. 1a, b) as potentially high-value targets, all of which were invariably downregulated in IBD-afflicted colons[9]. Reactome analyses showed that epithelial tight junctions (TJs), bioenergetics, and nuclear receptor pathway (PPAR signaling) related genes that are responsible for colon homeostasis are the major cellular processes regulated by these genes (Supplementary Fig. 1b). Downregulation of genes in clusters #1-3 was invariably associated also with an upregulation of genes in clusters #4-5-6; reactome analyses of the latter showed cellular processes that concern immune cell activation, inflammation, and fibrosis, which are hallmarks of IBD (Supplementary Fig. 1b). Of the druggable candidates within C#1-2-3, 17 targets were identified as associated with GO biological function of 'response to stress'/'response to stimuli'. Targeting one of the 17 targets, PRKAB1, the subunit of the heterotrimeric AMP-kinase engaged in cellular bioenergetics and stress response successfully restored the gut barrier function and also protected it from collapse in response to microbial challenge[9]. Here, we prioritized two more of those 17 targets, PPARA and PPARG, which encode a pair of nuclear receptors, PPARα and PPARγ, respectively. These two stress/stimuli-responsive genes are equivalent to each other and to PRKAB1, and like PRKAB1, are invariably downregulated in all IBD samples (Supplementary Fig. 1b–d). PPARA is in cluster #2 and PPARG is in cluster #3 (Supplementary Fig. 1b). They both were located on the two major Boolean paths associated with epithelial barrier and inflammation/fibrosis (Supplementary Fig. 1b)[9]. Together, these findings imply three things: (i) that PPARA and PPARG are simultaneously downregulated in IBD, (ii) that such downregulation is invariably associated with inflammation, fibrosis, and disruption of the epithelial barrier, and (iii) that simultaneous upregulation of PPARA and PPARG with agonists may restore the gut barrier. The last point is particularly important because Ppara/γ agonists are known to augment the expression of PPARA and PPARG, and depletion of either reduced the expression of the other[10].

Noteworthy, while the role of PPARγ in colitis has been investigated through numerous studies over the past 3 decades[11–13] (Supplementary Table 1), the role of PPARα has been contradictory (Supplementary Table 2), and their dual agonism in IBD has never been explored. The studies on PPARα are equally split on whether it is pro- or anti-inflammatory in

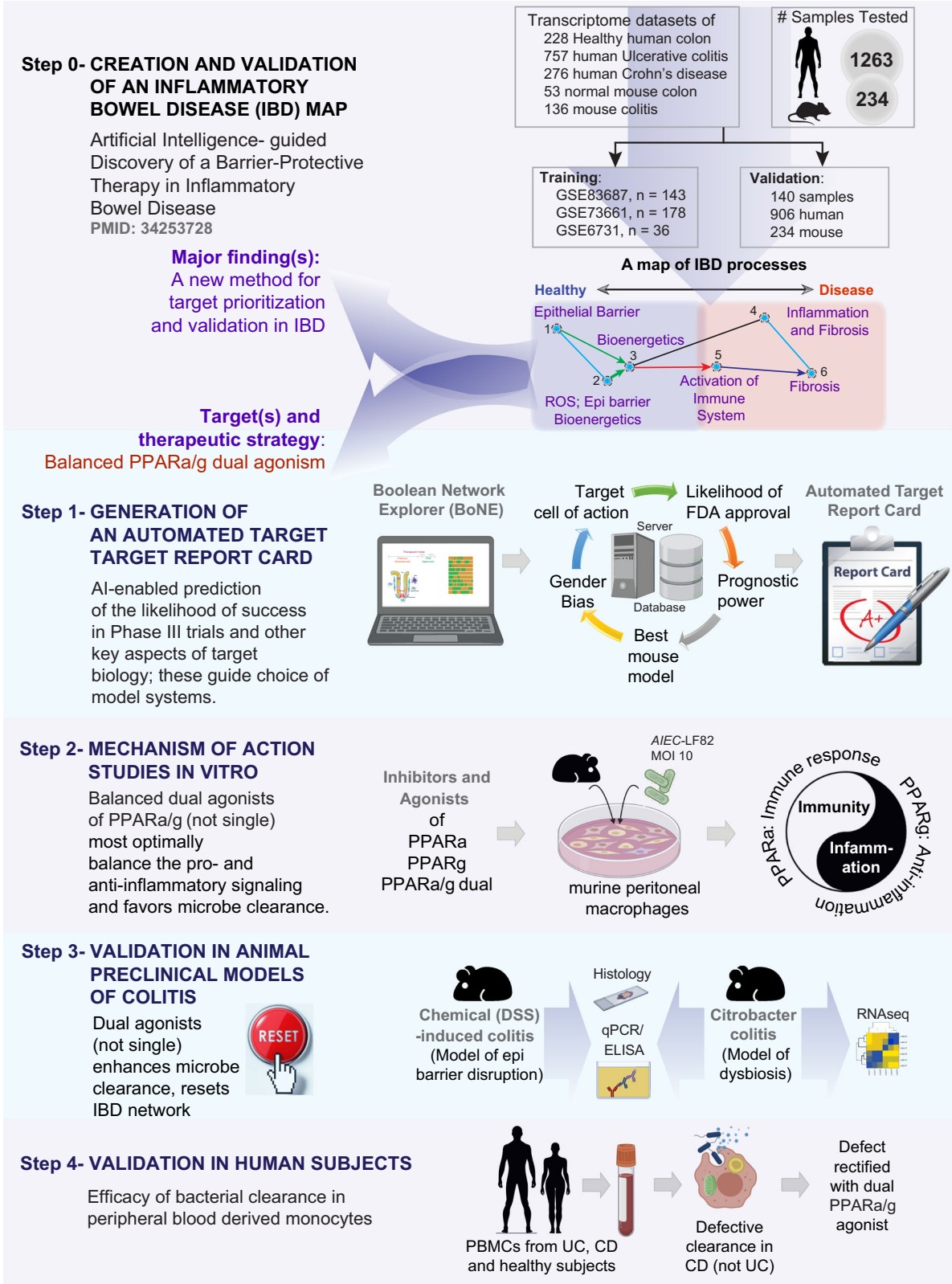

**Step 0- CREATION AND VALIDATION OF AN INFLAMMATORY BOWEL DISEASE (IBD) MAP**

Artificial Intelligence- guided Discovery of a Barrier-Protective Therapy in Inflammatory Bowel Disease
PMID: 34253728

**Major finding(s):**
A new method for target prioritization and validation in IBD

**Target(s) and therapeutic strategy:**
Balanced PPARa/g dual agonism

**Step 1- GENERATION OF AN AUTOMATED TARGET TARGET REPORT CARD**

AI-enabled prediction of the likelihood of success in Phase III trials and other key aspects of target biology; these guide choice of model systems.

**Step 2- MECHANISM OF ACTION STUDIES IN VITRO**

Balanced dual agonists of PPARa/g (not single) most optimally balance the pro- and anti-inflammatory signaling and favors microbe clearance.

**Step 3- VALIDATION IN ANIMAL PRECLINICAL MODELS OF COLITIS**

Dual agonists (not single) enhances microbe clearance, resets IBD network

**Step 4- VALIDATION IN HUMAN SUBJECTS**

Efficacy of bacterial clearance in peripheral blood derived monocytes

---

action[14–20]. By contrast, all studies on PPARγ agree that its agonism ameliorates DSS-induced colitis[13,21,22]. Although claimed to be effective on diverse cell types in the gut (epithelium, T-cells, and macrophages), the most notable target cells of PPARγ agonists are macrophages and dendritic cells[23]. Furthermore, Phase I and II clinical trials with PPARγ agonists either alone[24,25] or in combination with mesalamine[26] show barrier protective effects in UC patients. Despite these insights, the biopharmaceutical industry has not been able to harness the beneficial impact of this major target within emergent therapeutic strategies largely due to a trail of withdrawals after devastating long-term side effects including heart failure, bone fracture, bladder cancer, fluid retention, and weight gain[27,28]. Intriguingly, and of relevance to this work, the addition of PPARα agonistic activity to PPARγ,

**Fig. 1 Study design.** The premise of a 4-step drug discovery pipeline is summarized on the top (Step 0) is a recently published[9] Boolean implication network-based computational model of disease continuum states in inflammatory bowel disease (IBD map). The map, comprised of 6 gene clusters, was created and validated database containing 1497 gene–expression data (1263 human and 234 mouse samples). Paths, clusters and a list of genes in the network-based model were prioritized to discover one clinically actionable drug target (*PRKAB1*)[9]. Steps 1-4 outline the AI-guided identification and validation of another target pair, *PPARA*, and *PPARG*. Step 1: Dual agonists of PPARα/γ were predicted to—(i) modulate epithelial and macrophage processes; (ii) *Citrobacter* and chemical models of colitis were predicted as most optimal models; (iii) have high therapeutic index indicative of likelihood to succeed in Phase III clinical trials. Step 2: A combination of inhibitor and agonist studies helped establish that dual agonists reduce inflammation (PPARγ) while ensuring the induction of adequate immune response (PPARα). Step 3: Dural agonists ameliorated colitis in two preclinical models of colitis, and reversed the patterns of disease-associated gene expression that were altered in the IBD map. Step 4: In phase '0' human pre-clinical trials, PBMCs from CD, but not UC or healthy showed defective microbe clearance; this defect was reversed with a dual agonist of PPARα/γ.

PPARγ, to PPARδ agonists has led to a higher safety profile, leading to their development for use in many diseases, including type 2 diabetes, dyslipidemia, and non-alcoholic fatty liver disease[29].

**An automated target 'report card' for *PPARA* and *PPARG* in IBD**. We next generated an automated target report card for *PPARA* and *PPARG*. A high level of both PPARs, determined using a composite score for the abundance of both transcripts, was sufficient to distinguish healthy from IBD samples, not just in the test cohort that was used to build the IBD-map (ROC AUC of 0.74; Fig. 2b; see also Supplementary Fig. 2a–d), but also in four other independent cohorts with ROC AUC consistently above 0.88 (Fig. 2c). High levels of both PPARs also separated responders from non-responders receiving TNFα-neutralizing mAbs, GSE16879, E-MTAB-7604 or Vedolizumab that block the α4β7 integrin to prevent selective gut inflammatory, GSE73661 (ROC AUC 0.63-0.89, Fig. 2d), inactive disease from active disease (two independent cohorts ROC AUC above 0.93; Fig. 2d), and quiescent UC that progressed, or not to neoplasia (ROC AUC = 1.00 for qUC vs. nUC; Fig. 2d). A high level of *PPARA* and *PPARG* was also able to distinguish healthy from diseased samples in diverse murine models of colitis (Fig. 2e); but such separation was most effectively noted in some models (*Citrobacter* infection-induced colitis, adoptive T-cell transfer, TNBS, and *IL10*−/−), but not in others (DSS, and *TNFR1/2*−/−). These findings imply that therapeutics targeting these two genes are best evaluated in the murine models that show the most consistent decrease in the gene expression, e.g., *Citrobacter* infection-induced colitis, adoptive T-cell transfer, TNBS, etc. This was intriguing because the majority (~90%) of the published work on PPARα/γ dual agonists have been carried out in DSS models (Supplementary Tables 1, 2).

The expression profile of the target genes in the gut mucosa revealed that *PPARA* and *PPARG* are co-expressed at the highest levels in the crypt top epithelial cells and macrophages (Fig. 2f; Supplementary Fig. 3a–c), predicting that dual agonists are likely to preferentially act on these two cell types. The therapeutic index was below 0.1 for both genes (0.06 for *PPARA* and 0.04 for *PPARG*; Fig. 2g; Supplementary Fig. 2e, f), aligned well with two other FDA-approved targets shown on the line graph (ITGB1, 0.046 and JAK2, 0.032). The index, which is a statistical measure of the strength of association of Pparα/γ with genes that are targets of FDA-approved drugs that have successfully moved through the three phases of drug discovery (i.e., proven efficacy, with acceptable toxicity). A low number is indicative of a high likelihood of success in Phase-III trials. Finally, *PPARA* and *PPARG* expression was downregulated to a similar extent in men and women with IBD (Fig. 2h), predicting that therapeutics targeting them are likely to be effective in both genders.

**Rationalization of *PPARA/G* and *PPARGC1A* as targets in IBD**. Because proteins, but not transcripts, are the targets of therapeutic agents, the impact of therapeutics is translated to cellular processes via protein–protein interaction (PPI) networks, a.k.a interactomes. We next asked how dual agonists of PPARα/γ might impact cellular pathways and processes. A PPI network is visualized using PPARα and PPARγ as 'query/input' and the interactive STRING v11.0 database (https://string-db.org/) as a web resource of known and predicted protein–protein interactions curated from numerous sources, including experimental data, computational prediction methods, and public text collections. Pgc1a (a product of the gene *PPARGC1A*) was a common interactor between the two PPARs (Fig. 3a). We noted that Pgc1a also happens to be a major component within the PPARα/γ functional network, serving as a central hub for positive feedback loops between the PPARs and their biological function (Fig. 3b), i.e., mitochondrial biogenesis, DNA replication, and energetics (electron transport chain and oxidative phosphorylation). When we analyzed the functional role of the interactomes of PPARα/γ we noted that indeed both interactomes converged on lipid metabolism, mitochondrial bioenergetics and circadian processes (Fig. 3c, d), all representing major cellular processes that are known to be dysregulated in IBD[30–37]. These findings are consistent with the finding that *PPARA*, *PPARG,* and *PPARGC1A* are located within clusters #1-2-3 and all of them are predicted to be progressively and simultaneously downregulated in IBD samples (Fig. 3e; based on the IBD map, Supplementary Fig. 1).

**PPARA, PPARG, and PPARGC1A are downregulated in ulcerative colitis and Crohn's disease**. Previous work demonstrated that both PPARα and PPARγ are highly expressed in the colon[38] and that their expression (proteins and mRNA) is downregulated (by ~60%) in active UC[39], in both inflamed and noninflamed areas[40]. Moreover, the expression of PPARγ was significantly associated with disease activity[39]. Polymorphisms have also been detected in PPARγ; while some studies found those to be associated with an increased risk for CD[41], others found no evidence suggesting any form of association with an increased disease risk[42]. We collected endoscopically obtained biopsies from the colons of healthy ($n = 7$) and IBD ($n = 14$ and 14 of UC and CD, respectively) patients and assessed the levels of transcripts for *PPARA*, *PPARG*, and *PPARGC1A* by qPCR (Fig. 3f). We confirmed that all three transcripts were significantly downregulated in UC and CD samples compared to healthy; both *PPARG* and *PPARGC1a* were more significantly downregulated in CD compared to UC (Fig. 3g). These findings are in keeping with the network-based predictions that these genes should be downregulated invariably in all IBD samples, regardless of disease subtype (see individual disease maps; Supplementary Figs. 4, 5). While both *PPARA* and *PPARG* are in cluster #2 in the UC map, *PPARG* and P*PARA* are in separate clusters, clusters 2 and 6, respectively, in the CD map (Supplementary Figs. 4, 5). Reactome pathway analyses implied that in the case of UC, the two nuclear receptors may co-regulate similar cellular homeostatic processes associated with cluster #2, i.e., mitochondrial biogenesis and translation initiation,

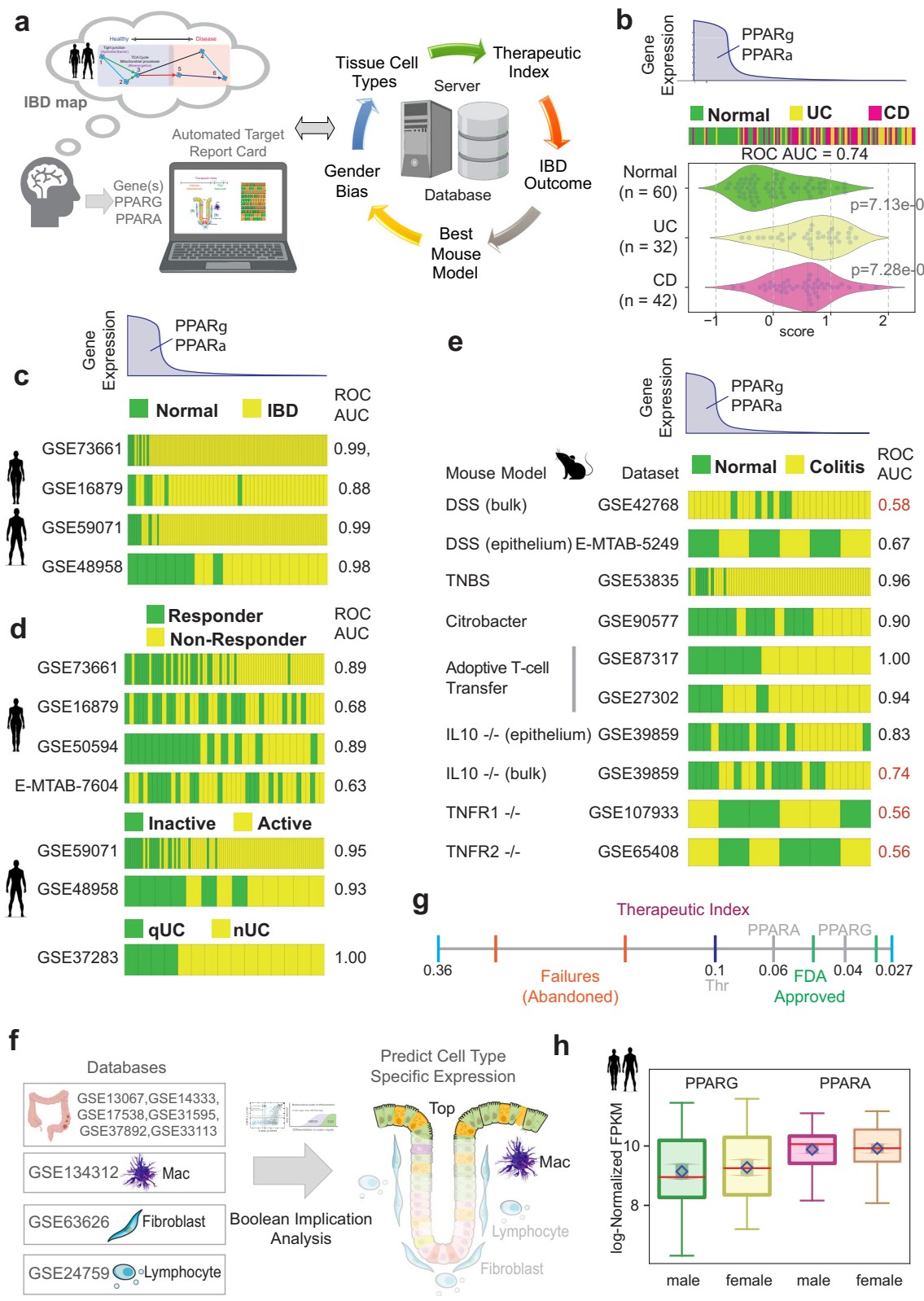

infectious disease, and detoxification of ROS (see Supplementary Fig. 4). By contrast, in the case of CD, they may independently regulate diverse cellular processes that maintain cellular homeostasis; while *PPARG* is associated with cellular metabolism (TCA cycle) and inhibition of NFkB signaling, *PPARA* is associated with transcriptional activity of nuclear receptors, cholesterol biosynthesis and Met/Ras signaling (see Supplementary

Fig. 5). Taken together, these findings demonstrate that *PPARA/ G* and *PPARGC1A* are downregulated in IBD and that they may regulate key pathophysiologic processes that are vital for cellular homeostasis. Findings support our AI-guided hypothesis that restoration of the expression of these genes will increase the expression of genes in C#1-2-3 and suppress the expression of, and that such increase.

**Fig. 2 Network-guided rationalization of PPARA/PPARG as targets in IBD. a** An interactive web-based platform allows the querying of paths of gene clusters in the IBD map [ref. [9]; see Supplementary Fig. 1] to pick high-value targets with a few mouse clicks and generate a comprehensive automated target 'report card'. The components of a 'target report card' is shown (right): predicted 'therapeutic index' (likelihood of Phase III success), IBD outcome (prognostic potential in UC and/or CD), network-prioritized mouse model, estimation of gender bias, and predicted tissue cell type of action. **b–h** Components of a target report card for *PPARA* and *PPARG* are displayed. Bar plot (b) displays the rank ordering of normal *vs.* ulcerative colitis (UC) /Crohn's Disease (CD) patient samples using the average gene expression patterns of the two genes: *PPARG/PPARA*. Samples are arranged from highest (left) to lowest (right) levels. ROC-AUC statistics were measured for determining the classification strength of normal vs IBD. Violin plots (**b**) display the differences in the average expression of the two genes in normal, UC, and CD samples in the test cohort that was used to build the IBD-map in[9]. Bar plots in panel **c–d** show the rank ordering of either normal *vs.* IBD samples (**c**) or responder vs. non-responder (R vs. NR; **d**), or active *vs.* inactive disease, or neoplastic progression in quiescent UC (qUC vs. nUC; **d**) across numerous cohorts based on gene expression patterns of *PPARG* and *PPARA*, from highest (left) to lowest (right) levels. Classification strength within each cohort is measured using ROC-AUC analyses. Bar plots in panel (**e**) show the rank ordering of either normal vs IBD samples across numerous published murine models of IBD based on gene expression patterns of *PPARG* and *PPARA* as in (**d**). ACT adoptive T cell transfer. Classification strength within each cohort is measured using ROC-AUC analyses. Bulk = whole distal colon; epithelium = sorted epithelial cells. Schematic in (**f**) summarizes the computational prediction of the cell type of action for potential *PPARA/G*-targeted therapy, as determined using Boolean implication analysis. GSEID# of multiple publicly available databases of the different cell types and colorectal datasets used to make sure predictions are cited. Red boxes/circles denote that *PPARA/G*-targeted therapeutics are predicted to work on monocytes/ macrophages and crypt-top enterocytes. Computationally generated therapeutic index (see Methods) is represented as a line graph in (**g**). The annotated numbers represent Boolean implication statistics. *PPARA* and *PPARG* align with other targets of FDA-approved drugs on the right of threshold (0.1). Two FDA-approved targets (green; *ITGB1*, 0.046; *JAK2*, 0.032), two abandoned targets (red; *SMAD7*, 0.33; IL11, 0.16), *PPARA* (gray, 0.064), *PPARG* (gray, 0.04), and the threshold (black, 0.1) are shown in the scale. Box plot in panel **h** shows that the level of *PPARA/G* expression is similar in the colons of both genders in health and in IBD, and hence, PPARα/γ-targeted therapeutics are predicted to have little/no gender predilection. The diamond square is the mean, and the arrows around it are 95% confidence interval.

## Synthesis and validation of PAR5359, a potent and specific PPARα/γ dual agonist.

We next sought to identify appropriate pharmacologic tools to test our hypothesis. Direct agonism of *PPARGC1A*/Pgc1a was deemed as not feasible because the only known agonist, ZLN005, non-specifically and potently also activates AMPK[43], a target that is known to independently improve barrier integrity in IBD[9]. Because Pgc1a is intricately regulated by feedback loops by PPARα/γ (Fig. 3b), we strategized targeting Pgc1a indirectly via PPARα/γ instead. As for PPARα/γ dual agonists, we noted that all commercially available compounds lack 'balanced' agonistic activities (Supplementary Table 3)[44,45]. Drugs that have fallen aside due to safety concerns also lack balanced agonism; most of them are more potent on PPARγ than on PPARα by a log-fold (Supplementary Table 3). All these PPARα/γ dual agonists have been withdrawn due to safety concerns[29], but the cause of the 'unsafe' profile remains poorly understood. Saroglitazar, the drug that is the only active ongoing Phase-III trial (NCT03061721) in this class, has ~3 log-fold more potency on PPARα than PPARγ[46]. Because our AI-guided approach suggested the use of simultaneous and balanced agonism, we favored the use of the only balanced and yet, specific PPARα/γ agonist described to date, PAR5359[47,48] (see Supplementary Table 4). In the absence of commercial sources or well-defined methods on how to synthesize this molecule, we generated PAR5359 in four synthetic steps (see details in Methods, (Supplementary Fig. 6)) and confirmed its specificity and the comparable agonistic activities using pure single PPARα [GW7647[49]] or PPARγ [Pioglitazone[50]] agonists as controls (Supplementary Fig. 7). With these potent and specific compounds as tools, and their doses adjusted to achieve the same potency, we set out to validate the network-based predictions using pre-clinical models.

## PAR5359 ameliorates *C. rodentium*-induced colitis, enhances bacterial clearance.

We next sought to assess the efficacy of individual and dual agonists of our compounds in murine pre-clinical models. PPARα/γ's role (or the role of their agonists) in protecting the gut barrier has been evaluated primarily in DSS-induced colitis (Supplementary Tables 1, 2). This model is more related to the UC patient pathology. However, *BoNE* prioritized other models over DSS, many of which accurately recapitulate the

PPARα and Pparγ-downregulation that is observed in the barrier-defect transcript signature in human IBD (Fig. 2e). Among those, we chose *C. rodentium*-induced infectious colitis, a robust model to study mucosal immune responses in the gut and understand derailed host-pathogen interaction and dysbiosis, which is closely related with IBD, and more specifically, CD pathophysiology[51–53]. This model is also known to emulate the bioenergetic dysbalance and mitochondrial dysfunction[54], both key cellular processes represented in C#1-2-3 within the IBD map (Supplementary Fig. 1). Furthermore, this model requires the balanced action of macrophages (a cell line predicted to be the preferred cell type target; Fig. 2f) to promote bacterial clearance and healing[55].

Colitis was induced by oral gavage of *C. rodentium* and mice were treated daily with the drugs via the intraperitoneal route (see Fig. 4a for workflow; Supplementary Fig. 8a). The dose for each drug was chosen based on their $EC_{50}$ on their respective targets so as to achieve equipotent agonistic activities (Supplementary Fig. 7). Fecal pellets of individual mice were collected to determine the number of live bacteria present in the stool. As anticipated, the bacterial burden in all mice increased from day 5, reaching a peak on day 7, forming a plateau until day 11 before returning to pre-infection baseline by day 18 (Fig. 4b). Compared against all other conditions, PAR5359-treated mice cleared the gut bacterial load significantly and rapidly (Fig. 4b–d). *Citrobacter* infection was associated with significant epithelial damage and profuse infiltration of inflammatory cells and edema by day 7 (Supplementary Fig. 8b) most of which resolved by day 18 (DMSO control; Fig. 4e). Colons collected on day 7 showed that treatment with PAR5359 significantly reduced these findings when compared to vehicle (DMSO), PPARα and PPARγ agonists alone (Supplementary Fig. 8b). Unexpectedly, when we analyzed the colons on day 18, we noted persistent immune infiltrates in tissues in two treatment arms, pioglitazone and GW7647 (arrowheads; Fig. 4e–j), but not in the vehicle control group, or those treated with PAR5359. These findings indicate that individual PPARα or PPARγ agonists may either retard bacterial clearance and/or induce an overzealous amount of inflammation, but the balanced-dual agonist (PAR5359) may have effectively cleared the infection and resolved inflammation. PAR5359 also reduced spleen inflammation as evidenced by a decreased spleen

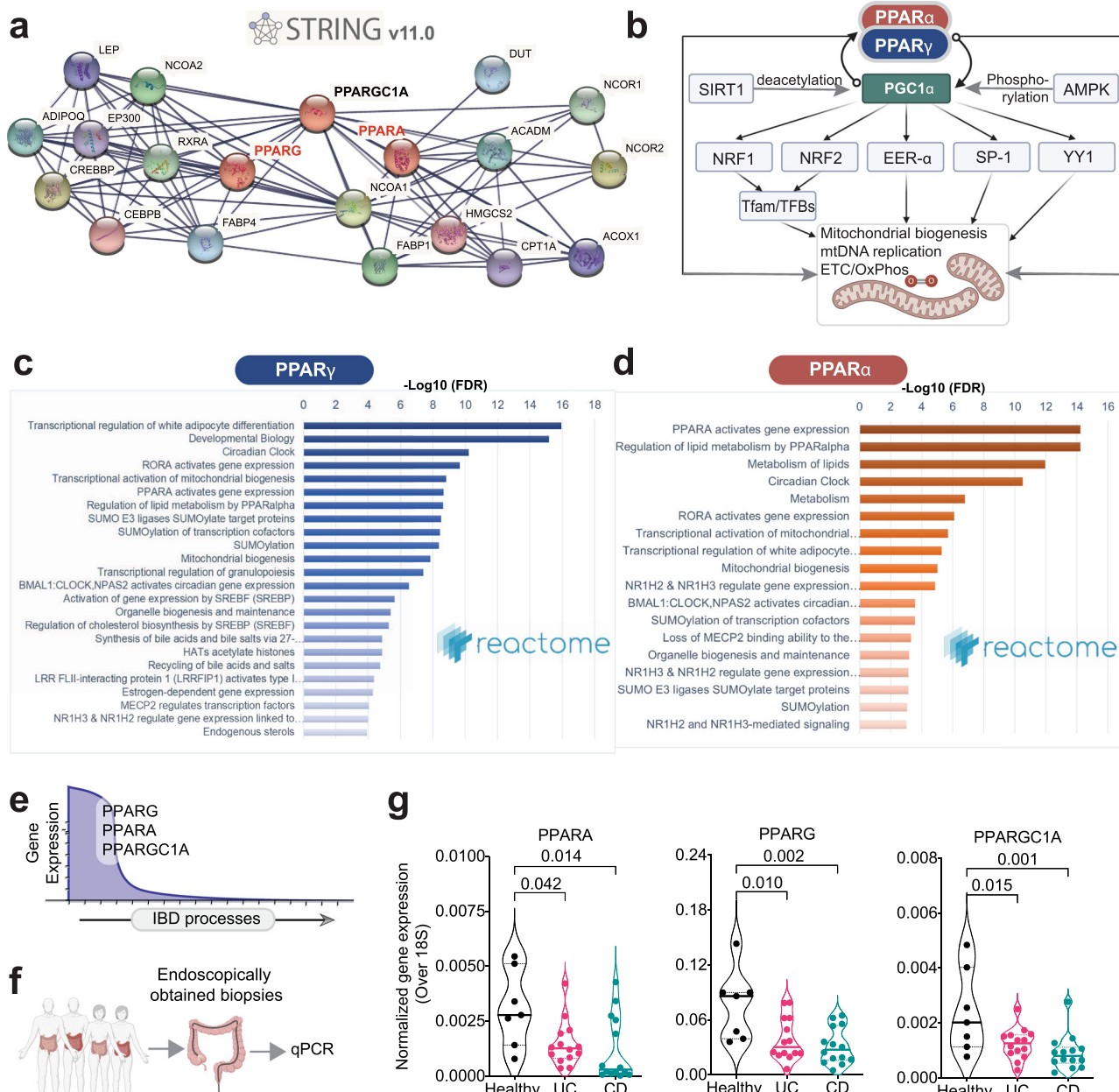

**Fig. 3 Rationalization of PPARα and PPARγ as targets in IBD. a** A protein–protein interaction network (i.e., interactomes) for PPARα and PPARγ, generated using STRING v.11 (https://string-db.org). **b** Schematic summarizing the roles of PPARα, PPARγ, and PGC1α on mitochondria biogenesis and function (based on). PGC1-α emerges as a critical hub for forward feedback loops. **c** Reactome pathway analyses (www.reactome.org) on PPARα and PPARγ interactomes in a show convergence on metabolism, mitochondria bioenergetics, and the circadian clock. **d** Graphical visualization of the predicted changes in the expression of three genes (and the proteins they encode): *PPARA* (PPARα), *PPARG* (PPARγ), and *PPARGC1A* (PGC1-α) during the progression of IBD processes (indicated with an arrow). **e**, **f** Schematic showing validation workflow; the expression of PPARA, PPARG, and PPARGC1A transcript levels were assessed in the ileum/colon biopsies of IBD patients (UC = 14 and CD = 14)) or healthy controls (n = 7). **g** Violin plots display the qPCR results in (**e**, **f**). Results are displayed as mean ± SEM. Significance was tested using one-way ANOVA followed by Tukey's test for multiple comparisons. Significance: n.s, not significant, *p*-value < 0.05 was considered significant.

weight and length compared to vehicle control (Supplementary Fig. 8c–f). The spleens of mice treated with DMSO, PPARα-alone agonist, GW7647, and PPARγ-alone agonist, Pioglitazone showed black-discoloration, presumably infarcts (arrows, Supplementary Fig. 8c, e). Notably, the spleens of mice treated with PPARα-alone agonist, GW7647, showed a significant increase in spleen length (Supplementary Fig. 8d, f).

Taken together, these findings indicate that PPARα/γ dual agonist PAR5359 is superior in ameliorating *C. rodentium*-induced colitis than either PPARα or PPARγ agonist used alone. Treatment with the dual, but not the single agonists hastened bacterial clearance, resolved inflammation, and induced healing.

**PAR5359 resets the colonic gene expression changes induced by *C. rodentium* infection**. Pharmacologic augmentation of *PPARA* and *PPARG* was hypothesized to be sufficient to upregulate genes in C#1-2-3, and restore the entire transcriptomic

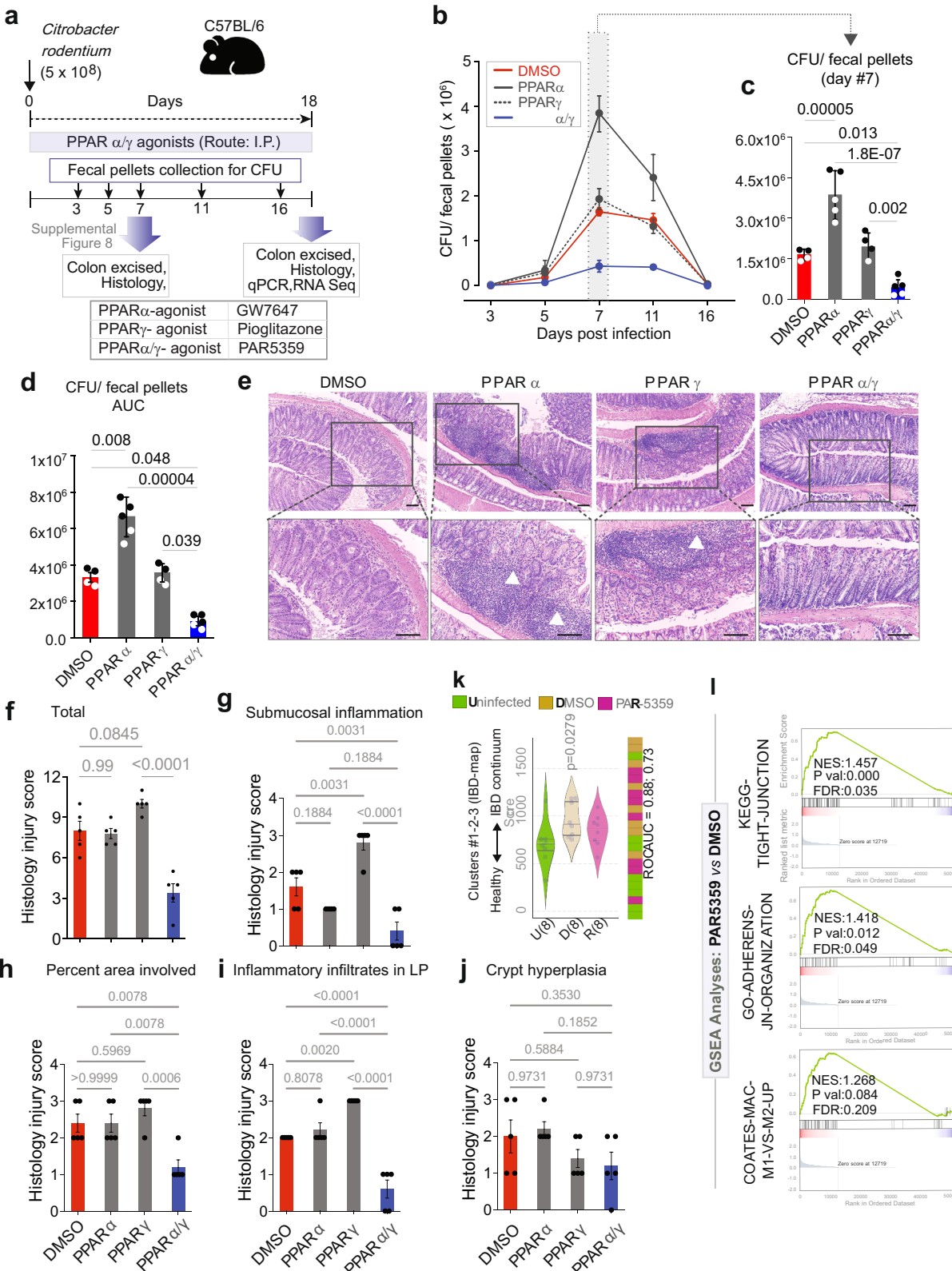

network to 'healthy' state via the invariant BIRs between the genes/clusters. We asked if that was achieved. RNA sequencing (RNA-seq) studies were carried out on the *C. rodentium*-infected colons in each treatment group (Fig. 4a). As expected, down-regulation of genes in clusters #1-2-3 of the IBD-map was sig-nificant in infected untreated (DMSO control) *vs.* uninfected

controls, indicative of network shift from health towards disease, and treatment with PAR5359 resisted such shift (Fig. 4k).

Pre-ranked gene set enrichment analyses (GSEA) based on pair-wise differential expression analysis showed that when compared to DMSO control, dual PPARα/γ agonism with PAR5359, but not individual agonists Pioglitazone or GW7647 was able to significantly preserve epithelial junction signatures

**Fig. 4 PPARα/γ dual agonists ameliorate *Citrobacter rodentium*-induced infectious colitis in mice. a** Schematic summarizing the workflow for testing PPARα/γ-targeted therapeutics in *C. rodentium*-induced colitis. Mice were gavaged with *C. rodentium* on day 0 and subsequently treated daily with PPAR agonists. Fecal pellets were collected to test viable bacterial burden, as determined by dilution plating and colony counting. Colons were excised on day 7 and 18 and analyzed using the indicated readouts. **b–d** Line graphs (**b**) display time series of the burden of viable bacteria in feces. Scatter plots with bar graphs (**c**) compare the peak burden of viable bacteria in feces on day 7. Scatter plots with bar graphs (**d**) display the area under the curve (AUC) for the line graph in b. CFU, colony forming units. Data points were plotted as black or white simply to improve visibility. **e** Images display representative fields from H&E-stained colon tissues of 4–5 mice in each group. Mag = 100x (top) and 200x (bottom); Scale bars, 100 μm. White arrowheads point to immune cell infiltrates. **f–j** Bar graphs display four parameters of inflammation that were quantified in H&E stained colon tissues in (**e**). Statistics: All results are displayed as mean ± SEM. Significance was tested on the cumulative histological score using one-way ANOVA followed by Tukey's test for multiple comparisons. Significance: n.s., not significant, *p*-value < 0.05 was considered significant. **k** Violin plots (left) display the deviation of expression of genes in Clusters #1-2-3 in the IBD network, as determined by RNA Seq on murine colons. Bar plot (right) displays the rank ordering of the samples. Welch's *t*-test was used to determine statistical significance. Significance: n.s., not significant, *p*-value < 0.05 was considered significant. **l** Pre-ranked GSEA based on pairwise differential expression analyses (DMSO vs PAR5359 groups) are displayed as enrichment plots for epithelial tight (top) and adherens (middle) junction signatures and balanced macrophage processes (bottom). See also Supplementary Fig. 8 for the day #7 results in the *C. rodentium*-induced colitis model, Supplementary Fig. 9 for extended GSEA analyses, and Supplementary Fig. 10 for the effect of PAR5359 on DSS-induced colitis in mice.

(both tight and adherens junctions) while maintaining balanced macrophage processes (compare Fig. 4l with Supplementary Fig. 9a). These enrichment analyses confirmed that, compared to DMSO-treated infected colons, the epithelial junction-related genes were significantly enriched in dual PPARα/γ-agonist treated samples (Fig. 4l, top and middle) without significant changes in macrophage polarization (Fig. 4l, bottom). These findings are in keeping with the predictions that epithelial cells and macrophages may be the primary cell type of action for dual PPARα/γ agonists. Comparison of all treatment cohorts against each other revealed that although both PAR5359 and Pioglitazone were superior to GW7647 in maintaining some epithelial processes (differentiation, tight junctions) and macrophage processes (Supplementary Fig. 9b–e), PAR5359 emerged as the only group that maintained homeostatic PPAR signaling in nature and extent as uninfected control (Supplementary Fig. 9f).

Taken together, these findings suggest that dual agonists of PPARα/γ are sufficient to either resist network shift and/or reverse the disease network in the setting of colitis. They also offer clues suggestive of epithelial and macrophage processes, two key cellular components of innate immunity in the gut lining as major mechanisms. These transcriptome-wide impacts suggest that PPARα/γ dual agonist PAR5359 is superior in restoring colon homeostasis in *C. rodentium*-induced colitis than either PPARα or PPARγ agonist used alone.

**PAR5359 ameliorates DSS-induced colitis.** It is well known that no single mouse model recapitulates *all* the multifaceted complexities of IBD[56,57]. Because almost all studies evaluating PPARα/γ-modulators have been performed on the DSS-induced colitis model (Supplementary Tables 1–2), we asked whether the PPARα/γ dual agonist PAR5359 can ameliorate colitis in this model. Mice receive intrarectal DMSO vehicle control or PAR5359 while receiving DSS in their drinking water (Supplementary Fig. 10a). Disease severity parameters, i.e., weight loss, disease activity index, shortening of the colon, and histology score were significantly ameliorated in the PAR5359-treated group (Supplementary Fig. 10a–e). These findings show that the PPARα/γ-dual agonist, PAR5359, is also effective in DSS-induced colitis. It is noteworthy that the PAR5359 dual agonist offered protection in the DSS-model, because prior studies using the same model have demonstrated that PPARα agonists worsen[16,18], and that the PPARγ agonists ameliorate colitis[58–60] (see Supplementary Tables 1, 2).

**PAR5359 promotes bacterial clearance with controlled production of ROS and inflammation in peritoneal macrophages.**

Next, we sought to study the mechanism of action of PAR5359, and the target cell type responsible for the superiority of dual agonism over single agonism. Our AI-guided approach predicted crypt top epithelium and macrophages as a site of action (Fig. 2f). Based on prior studies with single agonists in cell-specific KO mice (Supplementary Table 1) and the phenotypes observed in our animal models (Fig. 4; Supplementary Figs. 8–10), single PPARγ agonism appears sufficient to protect the epithelium in chemical-induced colitis (dual agonism did not offer additional advantage). The advantage of dual agonism is apparent in the *Citrobacter*-colitis model, which most robustly recapitulates the paradoxical immune suppression in the setting of dysbiosis that is seen in IBD, and most prominently in CD[51,61,62]. Because intestinal macrophages are alternatively polarized in this model[63,64], we hypothesized that balanced agonism may alter macrophage response to dysbiosis. To test this hypothesis, we incubated macrophages treated or not with the drugs and challenged them with CD-associated adherent invasive *E. coli* (AIEC)-LF82; this strain, originally isolated from a chronic ileal lesion from a CD patient[65]. As for the source of macrophages, we isolated metabolically active primary murine peritoneal macrophages using Brewer thioglycolate medium using established protocols[66,67] (Fig. 5a). These macrophages are known to have high phagocytic activity[66]. Thioglycolate-induced peritoneal macrophages (TG-PMs) were lysed, and viable intracellular bacteria were counted after plating on an agar plate. Pre-treatment with 1 μM PAR5359 and an equipotent amount of GW7647 (PPARα agonist) promoted bacterial clearance and reduced the bacterial burden when compared to vehicle control (Fig. 5b). By contrast, pre-treatment with Pioglitazone (PPARγ agonist) inhibited bacterial clearance; notably, the bacterial burden was significantly higher at both 3 and 6 h after infection (Fig. 5b). Reduced clearance of microbes in the latter was associated also with significant reductions in the cellular levels of reactive oxygen species (ROS) (Fig. 5c), which is believed to be a key component for effective bacterial killing[68]. By contrast, GW7647 did not reduce ROS induction, and PAR5359 treatment permitted an intermediate amount of induction of ROS (significantly lower than GW7647; Fig. 5c). Thus, the dual agonist PAR5359 allowed effective bacterial clearance (like GW7647; Fig. 5b) with minimal ROS induction (like Pioglitazone; Fig. 5c).

These patterns of microbial clearance and cellular ROS were associated also with the expression of cytokines, as determined by qRT-PCR analyses (Fig. 5d). As expected, infection of TG-PM with *AIEC*-LF82 induced *Il1β*, *Il6*, *Tnfα*, and *Il10*. PAR5359 significantly and selectively suppressed the expression of the pro-inflammatory cytokines *Il1β*, *Il6*, and *Tnfα* (but not the anti-inflammatory cytokine, *Il10*) (Fig. 5d). By contrast, the

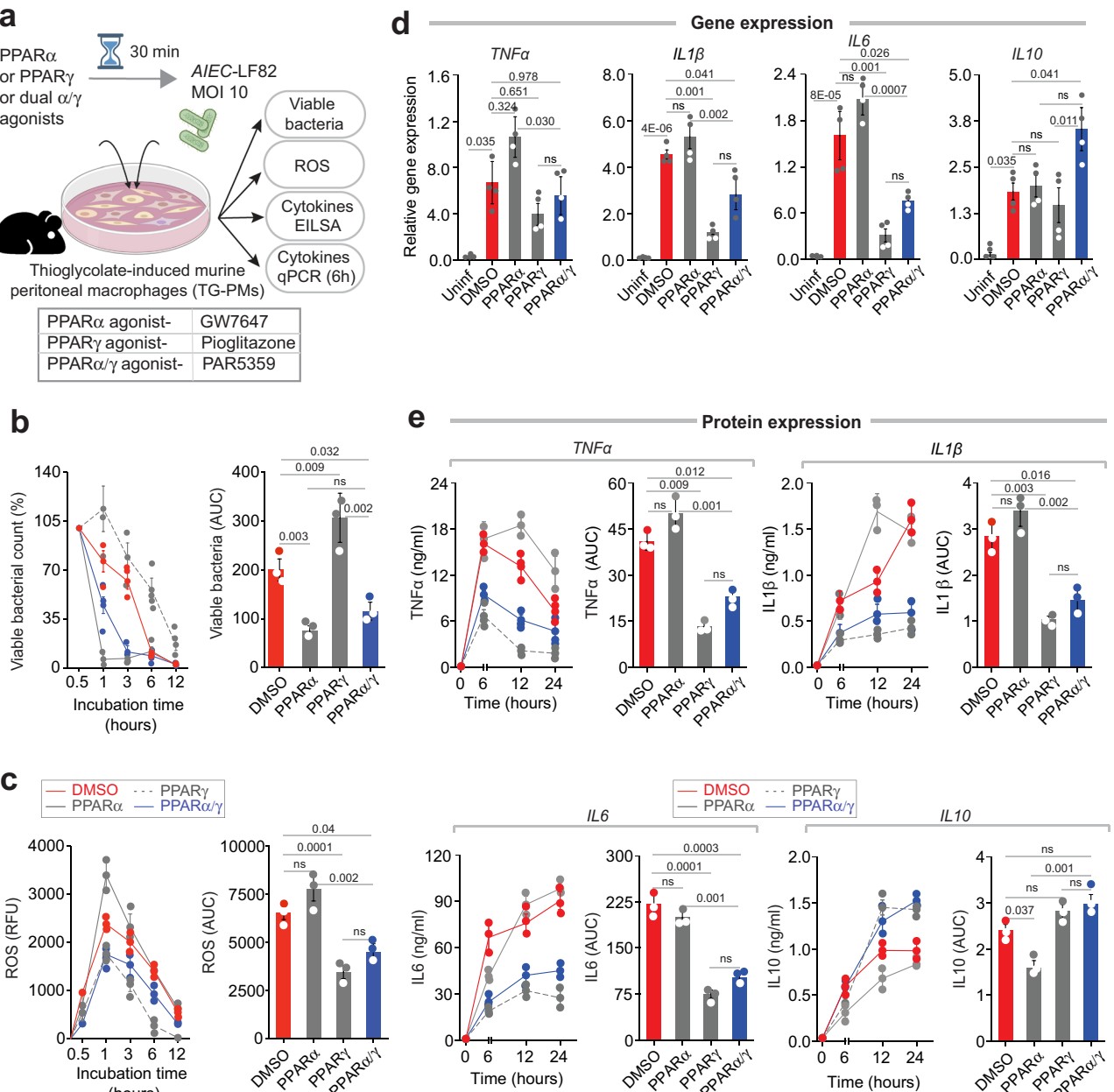

**Fig. 5 PPARα and PPARα/γ-dual agonists enhance, whereas PPARγ agonist delay bacterial (*AIEC*-LF82) clearance. a** Schematic displays the experimental design and workflow. Thioglycolate-induced murine peritoneal macrophages (TG-PM) pretreated with PPAR agonists (see box, below; 20 nM GW7647, 10 μM Pioglitazone and 1 μM PAR5359) were infected with *AIEC*-LF82 (*MOI* 10) and subsequently analyzed for the bacterial count (Gentamicin protection assay), generation of cellular ROS, secretion of inflammatory cytokines (in supernatant media by ELISA) and the induction of cytokines (gene transcript analysis by qPCR, the cycle threshold (Ct) of target genes was normalized to 18S rRNA gene and the fold change in the mRNA expression was determined using the $2^{-\Delta\Delta Ct}$ method). **b** Line graphs (left) display percent viable bacterial counts at indicated times after infection. Bar graphs (right) display the AUC. **c** Line graphs (left) and bar graphs (right) display the extent of ROS generation over time. **d** Bar graphs display the relative expression of transcripts of multiple cytokines (*Il1β, Il6, TNFα,* and *Il10*). All results are displayed as mean ± SEM. (*n* = 3). Significance: n.s, not significant, *p*-value < 0.05 was considered significant. **e** Line graphs (left) and bar graphs (right) showing the levels of indicated cytokines secreted in the media. Statistics: All results are from at least three independent experiments and results displayed as means ± SEM. Significance was tested using two-way/one-way ANOVA followed by Tukey's test for multiple comparisons. All results are displayed as mean ± SEM. (*n* = 3). Significance: ns, non-significant, *p*-value < 0.05 was considered significant. See Supplementary Fig. 11 for similar bacterial clearance assays performed using *Salmonella enterica*.

PPARγ specific agonist pioglitazone significantly suppressed the cytokines *Il1β* and *Il6*, and there was no effect of the PPARα specific agonist GW7647 (Fig. 5d). ELISA studies on the supernatant media further confirmed these findings (Fig. 5e), demonstrating that the effects in gene expression were also translated to the abundance of secreted cytokines released by the macrophages in the supernatant. Thus, both mRNA and protein estimations agreed that PPARα/γ dual agonism inhibits pro-inflammatory *Il1β* and *Il6* cytokines while permitting the induction of the major anti-inflammatory cytokine *Il10*. Similar findings were also observed in the case of another enteric pathogen, Salmonella (*S. enterica*); it is noteworthy that unlike

*AIEC*-LF82, *S. enteritica* is more efficient in surviving and multiplying inside the macrophage. The dual PAR5359 agonist enhanced bacterial clearance (Supplementary Fig. 11a, b) with limited induction of pro-inflammatory cytokines (significantly lower than control) and full augmentation of anti-inflammatory *Il10* cytokine (similar to control) (Supplementary Fig. 11c). Neither PPARα-, nor PPARγ- single agonists achieved this desirable profile, i.e., effective bacterial clearance with controlled inflammation. PPARα-agonist improved clearance, but failed to reduce proinflammatory cytokines. By contrast, PPARγ-agonist failed to promote clearance, but significantly reduced all cytokines (Supplementary Fig. 11).

Taken together, these studies on enteric pathogens show that-(i) PPARγ agonism induces 'tolerance' by suppressing inflammation, inhibiting ROS production and delaying bacterial clearance; (ii) PPARα agonism induces 'reactivity' by promoting bacterial clearance, permitting the full extent of ROS production and the induction of proinflammatory cytokines, but suppressing the anti-inflammatory il10 cytokine; and (iii) PPARα/γ dual agonism achieves a more balanced response; it suppresses proinflammatory cytokines without suppressing anti-inflammatory cytokine *Il10*, and thereby, permits the extent of inflammation and ROS induction that is optimal and sufficient to promote bacterial clearance.

**PPARα, but not PPARγ is required for the induction of inflammatory cytokines and ROS**. To further dissect which nuclear receptors are responsible for the balanced actions of the dual agonist, we next used a set of highly specific and potent PPARα/γ inhibitors (Supplementary Table 4). We pre-treated TG-PMs with PPARα and PPARγ inhibitors, either alone, or in combination, followed by stimulation with bacterial cell wall component LPS (Fig. 6a). As expected, LPS induced the cellular levels of ROS (Fig. 6b) and inflammatory cytokines (Fig. 6c, d) in TG-PMs significantly higher than in untreated control cells. Inhibition of PPARα suppressed the induction of cellular ROS and inflammatory cytokines, both at the level of gene and protein levels (Fig. 6b–d). By contrast, inhibition of PPARγ did not interfere with either response (Fig. 6b–d). Simultaneous inhibition of both PPARα and PPARγ mimicked the cellular phenotypes in the presence of PPARα inhibitors (Fig. 6b–d), indicating that inhibition of PPARα is sufficient to recapitulate the phenotype of dual inhibition. Taken together, these findings indicate that PPARα is required for the proinflammatory response of macrophages.

**PPARα/γ dual agonist PAR5359 promotes bacterial clearance in patient-derived PBMCs**. In search of a pre-clinical human model for testing drug efficacy, we next assessed microbial handling by PBMCs derived from patients with IBD and compared them with that in age-matched healthy volunteers. We enrolled both male and female patients and both CD and UC (Supplementary Table 5). Consecutive patients presenting for routine care to the UC San Diego IBD clinic were enrolled into the study; the only exclusion criteria were failure to obtain informed consent for the study or active infections and/or disease flare. Peripheral blood collected in the clinic was freshly processed as outlined in Fig. 7a to isolate PBMCs. Pre-treatment for 30 min with vehicle or PAR5359 was followed by infection for 1 h. Subsequently, the cells were treated with gentamicin for 60 min to kill extracellular bacteria to assess intracellular bacterial burden at 1 and 6 h after the gentamicin wash.

Two observations were made: First, CD but not UC patient-derived PBMCs when infected with *AIEC*-LF82 showed an increased number of internalized viable bacteria when compared to healthy PBMCs (Fig. 7b, e), indicative of either defective clearance and/or increased permissiveness to bacterial replication within the cells is limited to the CD. Second, pre-treatment with PAR5359 could improve clearance significantly (Figs. 7c, d, 7f, g). These results indicate that bacterial clearance is delayed in PBMCs of patients with CD and that PPARα/γ dual agonism with PAR5359 can reverse that defect. The possibility that such reversal could be due to any direct bacteriostatic/-cidal effect of PAR5359 agonist was ruled out (see bacterial viability assay in Supplementary Fig. 12). Our findings demonstrate that bacterial clearance is delayed primarily in CD and not UC are in keeping with the fact that delayed bacterial clearance from inflamed tissues (up to ~ 4-fold) is uniquely observed in CD[69]. These findings are also in keeping with our own observation that the downregulation of *PPARG/PPARGC1A* was more prominent in patients with CD (Fig. 3f, g). In fact, delayed clearance is one of the major reasons for persistent inflammation and disease progression among patients with CD[69,70].

## Discussion

Barrier-protection/restoration is the treatment endpoint for all clinical trials in IBD therapeutics; however, despite much success in the development of anti-inflammatory therapies[7,71], barrier-protective therapeutics in IBD have been slow to emerge[72]. Here we report the discovery of an effective barrier-protective therapeutic strategy in IBD identified using an AI-guided navigation framework (summarized in Fig. 8). First, a network-based drug discovery approach[9] was used to identify, rationalize and validate dual and balanced agonism of PPARα/γ (but not one at a time) is necessary for therapeutic success. Second, we provided evidence in the form of proof-of-concept studies (in two different pre-clinical murine models) demonstrating that the simultaneous and balanced agonistic activation of the pair of PPARs as an effective barrier protective strategy in IBD. Third, we demonstrate that macrophages are one of the primary target cell types of this therapeutic strategy; dual agonist (but not single) was permissive to the induction of macrophage responses expected for optimal immunity without overzealous inflammation. There are three notable takeaways from this study, which are unexpected observations and/or insights that fill key knowledge gaps in the fields of —(a) network medicine, (b) IBD therapeutics, and (c) macrophage biology.

First, with regard to network medicine, the AI-guided approach we used here differs from the current practice in three fundamental ways: (1) Unlike most studies that prioritize targets based on Differential Expression Analysis (DEA, or integrated DEA) or Bayesian approaches, target identification and prediction, this work was guided by a Boolean implication network (BIN) of continuum states in human disease[9]; (2) Instead of conventional approaches of trial-and-error, intuitive guess and/or knowledge-based prioritization of study models (animal or cell-type of action), target validation in network-rationalized animal and cell-type models that most accurately recapitulate the role of the target(s) during disease progression; (3) Inclusion of human pre-clinical model (patient-derived PBMCs) for target validation, inspiring the concept of Phase '0' trials that have the potential to personalize the choice of therapies. The combined synergy of these approaches validates a macrophage modulator in addressing the broken gut barrier in IBD.

The impact of using such an approach is fourfold: (i) Because the network approach used here relies on the fundamental invariant BIRs between genes, and their patterns of changes in expression between healthy and IBD samples, such 'rule of invariant' implies that any given relationship and/or change in expression pattern annotated within the network must be fulfilled

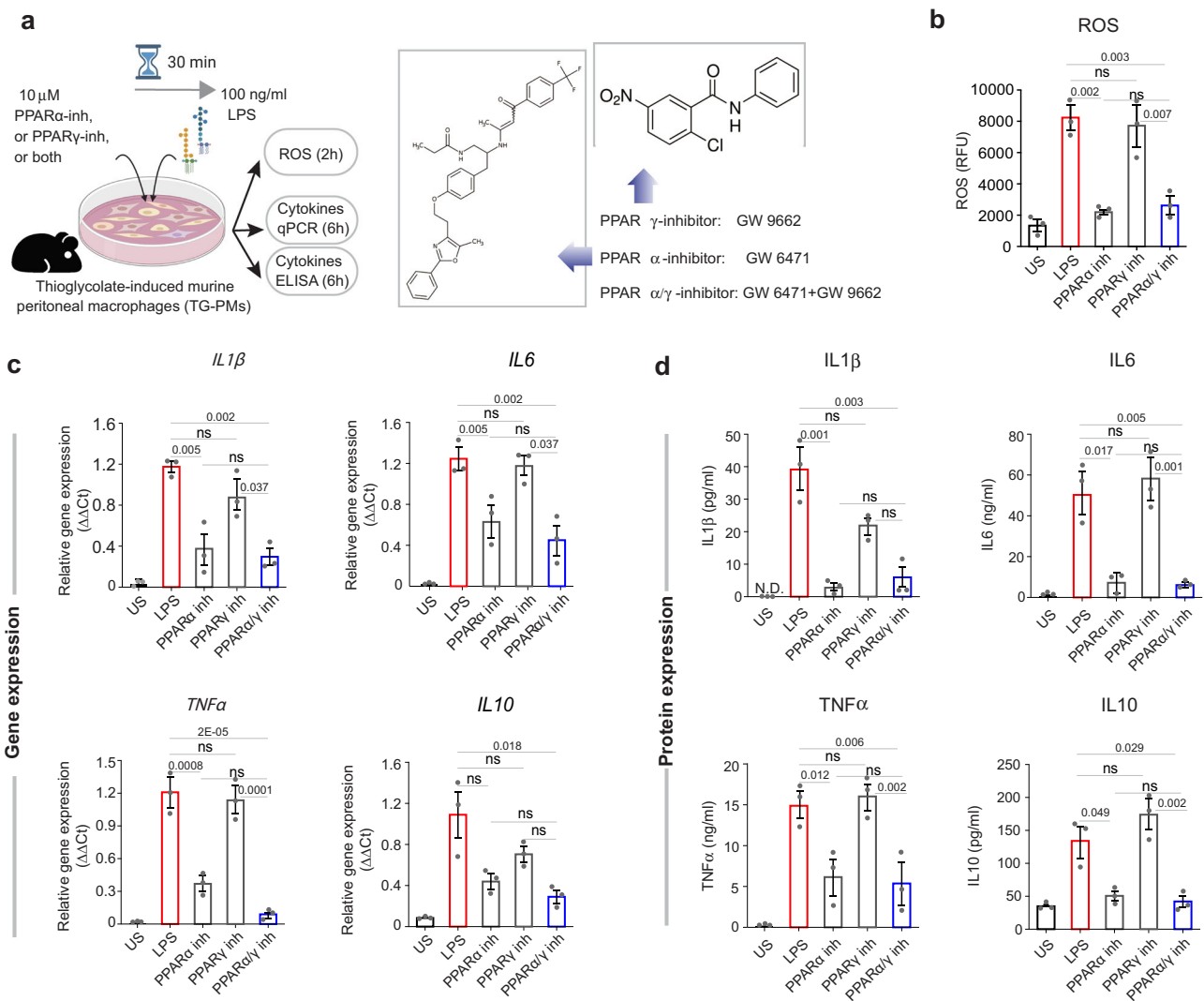

**Fig. 6 PPARα but not PPARγ is required for induction of cellular ROS and proinflammatory cytokines. a** Schematic of experimental design. TG-PMs were pre-incubated with 10 μM PPARα or PPARγ inhibitors (illustrated on the right), either alone or in combination for 30 min prior to stimulation with 100 ng/ml LPS. Cells were analyzed at 2 and 6 h to estimate cellular ROS and cytokine induction, respectively. **b–d** Bar graphs display the levels of cellular ROS (**b**), relative levels of mRNA (the target genes were normalized to 18S rRNA gene and the fold change in the mRNA expression was determined using the $2^{-\Delta\Delta Ct}$ method) (**c**) and protein (**d**) expression of cytokines (IL1β, IL6, TNFα, and IL10). Statistics: Results are from three independent experiments and displayed as mean ± SEM. One-way ANOVA followed by Tukey's test for multiple comparisons was performed to test significance. Significance: n.s, not significant, *p*-value < 0.05 was considered significant.

in every IBD patient. By that token, targets/drugs prioritized based on this network is expected to retain efficacy beyond inbred laboratory mice, into the heterogeneous patient cohorts in the clinic. (ii) This AI-guided approach not just helped compute pre-test probabilities of success ("Therapeutic Index"), but also helped pick models that are most insightful and appropriate to demonstrate therapeutic efficacy (e.g., *Citrobacter rodentium* infection-induced colitis) and to pinpoint the cell type and mechanism of action (microbial clearance by macrophages). This is noteworthy because the conventional approach in studying PPARs has been limited to the use of DSS-induced colitis (see Supplementary Tables 1, 2), which has often given conflicting results (see Supplementary Table 2). PPARγ agonists work best for the UC patients, perhaps because it is a potent inhibitor of proinflammatory cytokines and, as shown before, protects the intestinal epithelium[38]. Our findings in the *Citrobacter* model imply that such single PPAR γ agonism may worsen the macrophage dysfunction that is observed in the setting of CD, which is

characterized by ineffective microbial clearance, insufficient proinflammatory response in the setting of luminal dysbiosis[35]. In fact, without the use of the *Citrobacter rodentium* infectious colitis model, the deleterious effects of PPARγ agonists would have been overlooked. (iii) Having a computational framework improves precision in target identification; it is because of the emergence of the two PPARs (alongside their positive feedback regulator, Pgc1a) within our network, we rationalized their dual agonism as a preferred strategy (over single) and our experiments validated that prediction both in vivo and in vitro. This is noteworthy because conventional approaches have demonstrated a protective role of PPARγ agonists and a conflicting (both protective and exacerbating) role of PPARα in IBD;[15,17,18,73] the advantage of dual agonism has neither been rationalized nor tested. (iv) The 'target report card', like the one shown here, is a project navigation tool that is geared to streamline decision-making (i.e., which genes, which animal models, which cell type/cellular process, what is the likelihood of success, etc.), which in

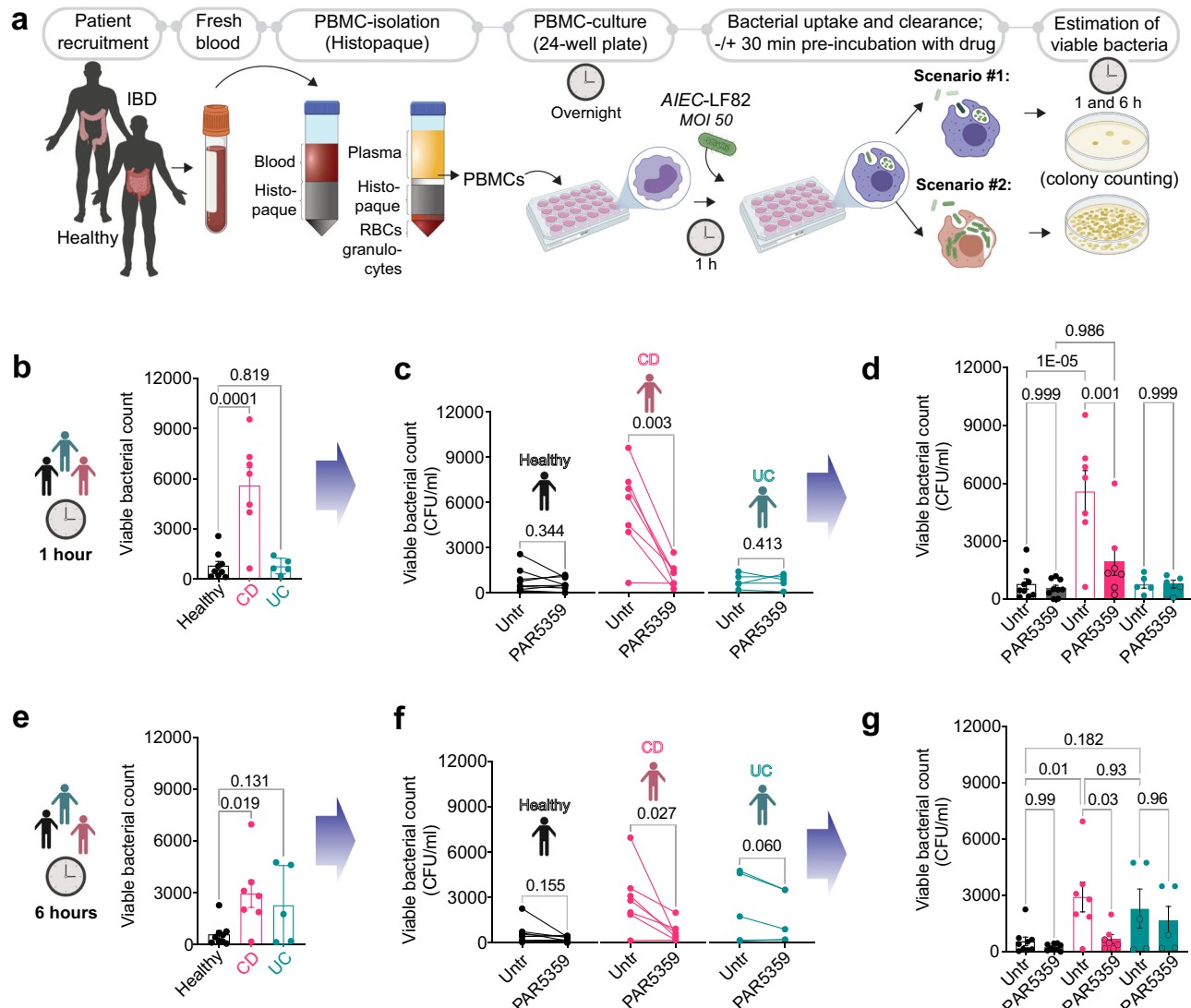

**Fig. 7 PPARα/γ dual agonist, PAR5359, promotes the clearance of *AIEC*-LF82 from CD patient-derived PBMCs. a** Schematic displays the overall experimental design using human subjects (see Supplementary Table 5 for patient demographics). Peripheral blood collected from healthy, CD, and UC patients was used as a source of PBMCs. PBMCs were pre-treated for 30 min with 1 μM PPARα/γ agonists prior to infection with *AIEC*-LF82 (*MOI* 50) for 1 h. PBMCs were subsequently treated with gentamicin to kill extracellular microbes for 60 min (~t0 h) prior to lysis and plating to determine the intracellular abundance of viable bacteria at 1 and 6 h, as determined by dilution plating and colony counts (see Methods for details). **b–g** Bar graphs with scatter plots display the abundance of viable intracellular bacteria at 1 h (**b**) and 6 h (**e**) after infection. Paired line plots display the rate of clearance of bacteria in individual subjects at 1 h (**c**) and 6 h (**f**) after infection. Data in (**b**, **c**) of 1 h infection is combined in (**d**) and data in (**e**, **f**) of 6 h infection is combined in (**g**) with statistics: Results are displayed as mean ± SEM (CD patient $n = 7$, UC patients = 6, and healthy $n = 9$). Paired *t*-test or one-way ANOVA followed by Tukey's test for multiple comparisons was performed to test significance. Significance: n.s., not significant, *p*-value < 0.05 was considered significant.

turn should reduce attrition rates, waste, and delays; the latter are well-recognized flaws in the current process of drug discovery.

Second, regarding IBD therapeutics, our studies demonstrate that single or unbalanced combinations of Ppar agonists are inferior to dual/balanced agonists. Conventional and reductionist approaches have inspired numerous studies with single Ppar agonists over the past decade (Supplementary Tables 1, 2). However, given the devastating side effects of most single or unbalanced PPARα/γ agonists (Supplementary Table 3), translating to the clinic beyond a Phase II trial[24,74,75] has not been realized. Because the therapeutic index for the dual PPARα/γ agonists matches that of other FDA-approved targets/drugs, it is predicted that barring unexpected side effects, dual Ppar agonists are likely to be effective as barrier-protective agents. As for side

effects, we noted is that balanced PPARα/γ agonists are rare; while all dual PPARα/γ agonists that have been discontinued due to side effects happen to be either single (only PPARγ) or 'unbalanced' (PPARγ » PPARα agonistic activity), the newer generation formulations that are currently in the clinical trial have a reversed agonistic potency (PPARα » PPARγ agonistic activity) (see Supplementary Table 3). Because macrophage responses require finetuning (discussed below), our studies show how unopposed agonism of either PPARγ or PPARα is harmful and can impair/dysregulate the way macrophages respond when microbes breach past the gut barrier. It is possible that many of the side effects of the discontinued thiazolidinediones are due to their inability to achieve that 'optimal' spectrum of macrophage function.

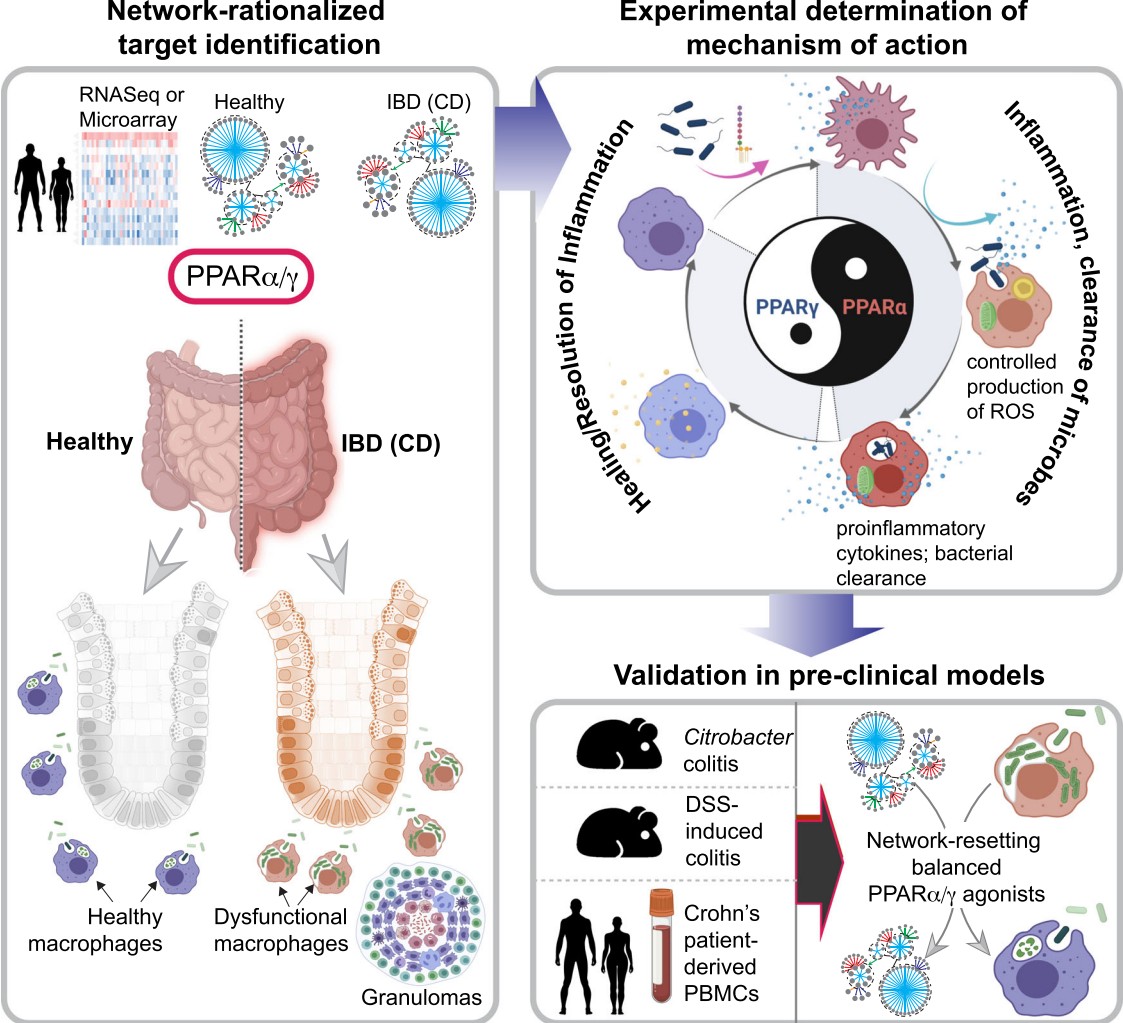

**Fig. 8 Summary of findings and working model.** Schematic summarizes key approaches and findings of this study. First, network-rationalized target identification (Left) was performed using a web-based platform that queries > 1000 IBD datasets [ref. 9; see Methods] that served as 'input' to create a map of gene clusters that are progressively altered in the gut in the setting of IBD. Predictions are used to guide the choice of therapeutics (dual agonists of PPARα and PPARγ that have a balanced agonistic potential for both PPARs), the choice of animal models of IBD, predict cell types of action (macrophage processes), and finally, the subtype of IBD that could benefit most based on the cell type of action (i.e., CD). Second, experimentally determined mechanism of action studies (right, top) showed that balanced actions of both PPARα and PPPARγ enable the induction of bacterial clearance, resolution of inflammation, and healing; PPARα is responsible for ROS and cytokine induction, whereas PPPARγ is responsible for anti-inflammatory response and healing. The dual agonistic action was superior to each agonist used alone. Third, targets validation studies (right, bottom) in murine and human models confirm the use of PPARα/γ dual agonists for enhancing bacterial clearance and protection against colitis. When tested side-by-side in the infectious colitis model, the dual agonistic action was superior to each agonist used alone.

Third, when it comes to macrophage biology, this work sheds some unexpected and previously unforeseen insights into the role of the PPARs in the regulation of macrophage processes. Extensively studied for over ~3 decades, PPARs are known to regulate macrophage activation in health and disease[76]. Targeting PPARs as a host-directed treatment approach to infectious/inflammatory diseases appears to be a sound strategy because they regulate macrophage lipid metabolism, cholesterol efflux, inflammatory responses (ROS and cytokine production), apoptosis, and production of antimicrobial byproducts[77]. We found that unopposed PPARγ activation suppresses bacterial clearance and blunts the induction of proinflammatory (but not anti-inflammatory, IL10) cytokines and ROS in response to infection both in vivo and in vitro. In other words, and consistent with prior reports, PPARγ activation suppressed inflammation at the cost of impairing immunity. Our findings are in keeping with the findings of a systematic review and meta-analysis of 13 long-term randomized

controlled trials that involved 17,627 participants (8,163 receiving PPARγ agonists and 9464 receiving control drugs)[78]. Long-term (~1–5.5 y) use of PPARγ agonists increases the risk of pneumonia or lower respiratory tract infection significantly, some of which result in hospitalization, disability, or death[78]. In the case of PPARα, unopposed activation-induced ROS and proinflammatory cytokines and accelerated bacterial clearance. Inhibitor studies further confirmed that PPARα was required for these responses (Fig. 6). These findings are in keeping with others' showing that PPARα, but not PPARγ is required for NADPH-induced ROS formation both in human and murine macrophages[79]. PPARα agonists induce the expression of NADPH oxidase subunits p47(phox), p67phox, and gp91phox, which are all essential functional components of NADPH complex[79]. Dual and balanced PPARα/γ agonism enhanced bacterial clearance with only a moderate induction of proinflammatory cytokines or ROS. Such a response ensures that the

macrophage functions within a 'goldilocks' zone, mounting inflammation that is just sufficient for microbial clearance and immunity[80]. In our analysis, the only other PPARrelated gene within the IBD network, i.e., Pgc1a, and its role within the PPARα/γ axis suggests that the intricate network of forward feedback loops orchestrated by Pgc1a may be critical for achieving the critical balance between immunity and inflammation, which is a key outcome of the dual PPARα/γ agonists.

Because previous studies using cell-specific gene depletion have indicated that the barrier-protective role of PPARγ may be mediated via cells other than the macrophages[59], namely, the T cells[81] and the epithelial cells[38], it is possible that the dual PPARα/γ agonists also act on those cells, promoting bacterial clearance and balancing cellular bioenergetics, ROS and cytokine production, in manners similar to that we observe in macrophages.

Taken together, our study uses an unconventional approach to rationalize and validate the use of PPARα/γ dual agonists as barrier protective macrophage modulators in the management of IBD. The approach is powerful because it leverages the precision of mathematics (Boolean algebra of logic) and the fundamental invariant patterns in gene expression (Boolean Implications). The AI-navigated drug discovery approach defined here could serve as a blueprint for future studies not just in IBD, but in any other such complex chronic diseases.

## Method

The identity and source of all resources (reagents, computational and other software and equipment) used for this study is cataloged in Supplementary Table 6.

### Computational

*A Boolean network map of IBD.* A BIN was created earlier [ref. [9]; Supplementary Fig. 1a], and this network is comprised of clusters of genes, interconnected by BIRs. The concepts, mathematical, statistical, datasets that went into building this map is detailed in ref. [9], and briefly mentioned here.

*Gene expression databases.* Publicly available human colon tissue gene expression databases were downloaded from the National Center for Biotechnology Information (NCBI) Gene Expression Omnibus website (GEO)[82–84]. If the dataset is not normalized, RMA (Robust Multichip Average)[85,86] is used for microarrays and TPM (Transcripts Per Millions)[87,88] is used for RNASeq data for normalization. We used log2(TPM + 1) to compute the final log-reduced expression values for RNASeq data. Accession numbers for these crowdsourced datasets are provided in the figures and manuscript. All of the above datasets were processed using the Hegemon data analysis framework[89–91].

*Boolean analysis.* Boolean logic is a simple mathematic relationship of two values, i.e., high/low, 1/0, or positive/negative. The Boolean analysis of gene expression data requires first the conversion of expression levels into two possible values. The StepMiner algorithm is reused to perform Boolean analysis of gene expression data[92]. The Boolean analysis is a statistical approach that creates binary logical inferences that explain the relationships between phenomena. The Boolean analysis is performed to determine the relationship between the expression levels of pairs of genes. The StepMiner algorithm is applied to gene expression levels to convert them into Boolean values (high and low). In this algorithm, first, the expression values are sorted from low to high and a rising step function is fitted to the series to identify the threshold. The middle of the step is used as the StepMiner threshold. This threshold is used to convert gene expression values into Boolean values. A noise margin of 2-fold change is applied around the threshold to determine intermediate values, and these values are ignored during Boolean analysis. In a scatter plot, there are four possible quadrants based on Boolean values (Supplementary Fig. 2a): (low, low), (low, high), (high, low), (high, high).

*Invariant Boolean implication relationships.* A BIR is observed if any one of the four possible quadrants or two diagonally opposite quadrants are sparsely populated. Based on this rule, there are six different kinds of BIRs. Two of them are symmetric: equivalent (corresponding to the highly positively correlated genes), opposite (corresponding to the highly negatively correlated genes). Four of the Boolean relationships are asymmetric, and each corresponds to one sparse quadrant: (low => low), (high => low), (low => high), (high => high). BooleanNet statistics (Equations listed below) is used to assess the sparsity of a quadrant and the significance of the BIRs[92,93]. Given a pair of genes A and B, four quadrants are identified by using the StepMiner thresholds on A and B by ignoring the

Intermediate values defined by the noise margin of twofold change (±0.5 around StepMiner threshold). The number of samples in each quadrant is defined as $a_{00}$, $a_{01}$, $a_{10}$, and $a_{11}$ (Supplementary Fig. 2a). The total number of samples where gene expression values for A and B are low is computed using the following equations.

$$nA_{low} = (a_{00} + a_{01}), nB_{low} = (a_{00} + a_{10}) \tag{1}$$

Total number of samples considered is computed using the following equation.

$$total = a_{00} + a_{01} + a_{10} + a_{11} \tag{2}$$

The expected number of samples in each quadrant is computed by assuming independence between A and B. For example, the expected number of samples in the bottom left quadrant $e_{00} = \hat{n}$ is computed as probability of A low $((a_{00} + a_{01})/total)$ multiplied by the probability of B low $((a_{00} + a_{10})/total)$ multiplied by the total number of samples. The following equation is used to compute the expected number of samples.

$$n = a_{ij}, \hat{n} = (nA_{low}/total * nB_{low}/total) * total. \tag{3}$$

To check whether a quadrant is sparse, a statistical test for $(e_{00} > a_{00})$ or $(\hat{n} > n)$ is performed by computing $S_{00}$ and $p_{00}$ using the following equations. A quadrant is considered sparse if $S_{00}$ is high $(\hat{n} > n)$ and $p_{00}$ is small.

$$S_{ij} = \frac{\hat{n} - n}{\sqrt{\hat{n}}}$$

$$p_{00} = \frac{1}{2}\left(\frac{a_{00}}{(a_{00} + a_{01})} + \frac{a_{00}}{(a_{00} + a_{10})}\right) \tag{4}$$

A threshold of $S_{00} > sthr$ and $p_{00} < pthr$ to check sparse quadrant. A BIR is identified when a sparse quadrant is discovered using the following equation.

$$Boolean\ Implication = (S_{ij} > sthr, p_{ij} < pthr)$$

A relationship is called Boolean equivalent if top-left and bottom-right quadrants are sparse (Supplementary Fig. 2b).

$$Equivalent = (S_{01} > sthr, P_{01} < pthr, S_{10} > sthr, P_{10} < pthr) \tag{5}$$

Boolean opposite relationships have sparse top-right $(a_{11})$ and bottom-left $(a_{00})$ quadrants.

$$Opposite = (S_{00} > sthr, P_{00} < pthr, S_{11} > sthr, P_{11} < pthr) \tag{6}$$

Boolean equivalent and opposite are symmetric relationship because the relationship from A to B is same as from B to A. Asymmetric relationship forms when there is only one quadrant sparse (A low => B low: top-left; A low => B high: bottom-left; A high => B high: bottom-right; A high => B low: top-right). These relationships are asymmetric because the relationship from A to B is different from B to A. For example, A low => B low and B low => A low are two different relationships.

A low => B high is discovered if bottom-left $(a_{00})$ quadrant is sparse and this relationship satisfies the following conditions.

$$A\ low => B\ high = (S_{00} > sthr, P_{00} < pthr) \tag{7}$$

Similarly, A low => B low is identified if top-left $(a_{01})$ quadrant is sparse.

$$A\ low => B\ low = (S_{01} > sthr, P_{01} < pthr) \tag{8}$$

A high => B high Boolean implication is established if the bottom-right $(a_{10})$ quadrant is sparse as described below.

$$A\ high => B\ high = (S_{10} > sthr, P_{10} < pthr) \tag{9}$$

Boolean implication A high => B low is found if top-right $(a_{11})$ quadrant is sparse using the following equation.

$$A\ high => B\ low = (S_{11} > sthr, P_{11} < pthr) \tag{10}$$

For each quadrant, a statistic $S_{ij}$ and an error rate $p_{ij}$ is computed. $S_{ij} > 2.5$ and $p_{ij} < 0.1$ are the thresholds used on the BooleanNet statistics to identify BIRs. The false discovery rate is computed by randomly shuffling each gene and computing the ratio of the number of BIR discovered in the randomized dataset and original dataset. For IBD dataset the false discovery rate was <0.001.

Boolean Implication analysis looks for invariant relationship across all the different types of samples regardless of the conditions and treatment protocols. Therefore, it does not distinguish the sample types when discovering BIRs (Supplementary Fig. 2c, d). We assume that there are fundamental invariant Boolean implication formula that is satisfied by every sample regardless of their type (in this context it is limited to healthy and IBD colonic biopsies including both UC and CD). This means normal, UC, and CD samples share the same fundamental relationships.

*IBD datasets.* Both Peters-2017 GSE83687 and Arijs-2018 GSE73661 dataset were independently prepared for Boolean analysis by filtering genes that have reasonable dynamic range of expression values by analyzing the fraction of high and low values identified by the StepMiner algorithm[94]. Any probeset or genes that contain <5% of high or low values or do not have a big dynamic range are dropped from the analysis (for Peters-2017 dataset 7659/23228 genes dropped—33%). To check if pairwise BIRs are consistent between two datasets, every gene in Peters-2017 dataset is mapped to the best probeset (identified by the biggest dynamic range) in

the Arijs-2018 dataset, and genes/probesets that do not match are dropped from the analysis (4841/23228 genes dropped—21%). Finally, 44% (10232/23228) of genes were not used in the Boolean Implication Network because their expression did not have a sufficient range. Since RNA-Seq expression values have slightly different characteristics than microarray expression values, the consistency of BIR was determined by using BooleanNet statistics in both datasets and a Pearson's correlation coefficient in the Arijs-2018 dataset. A Pearson's correlation coefficient > 0.5 was considered compatible with Equivalent, High => High, and Low => Low BIRs. Similarly, a Pearson's correlation coefficient < −0.25 was considered compatible with Opposite, High => Low, and Low => High BIRs. The Boolean model is tested in several human datasets, each comprised of a heterogeneous collection of samples to demonstrate reproducibility (GSE16879, GSE59071, GSE48958, GSE50594, GSE37283, E-MTAB-7604). We have collected publicly available gene expression datasets derived from mouse models of IBD (DSS bulk GSE42768, DSS epithelium E-MTAB-5249, TNBS GSE53835, Citrobacter GSE90577, adoptive T-cell transfer ACT GSE87317, ACT GSE27302, IL10 -/- GSE39859, TNFR1 -/- GSE107933, TNFR2 -/- GSE65408) to test whether human Boolean models performs well in mice. The gene name conversion from human to mouse is performed using human genome GRCh38.95 ensembl IDs and mapping data exported from ensemble BioMart web-interface.

*Generation of target report card.* A target report card is generated for one target or multiple targets to predict the efficacy of a potential drug. The target report card contains five different sections as described below:[1] Therapeutic index[2], IBD outcome[3], Network-prioritized mouse model[4], estimation of gender bias[5], Predicted tissue cell type of action.

*Target report card – Therapeutic index.* Therapeutic index is a number (lower the better) assigned to one target or multiple targets that predicts the efficacy of a potential drug. The therapeutic index is computed by measuring the strength of BIR with PRKAB1 (Supplementary Fig. 2e). Since PRKAB1 is an agonist, if $S_{11} > 0$ then gene A is an Antagonist because top-right quadrant will have fewer samples than expected and PRKAB1 high will be associated with gene A low. For antagonist, top-right and bottom-left quadrants are expected to be sparse. Therefore, $T_{index}$ for antagonist is computed as follows:

$$T_{index} = \frac{1}{4}\left(\frac{0.3}{(S_{00}+1)} + \frac{0.3}{(S_{11}+1)} + p_{00} + p_{11}\right) \quad (11)$$

Similarly, if $S_{11} > 0$ then gene A is an Agonist because top-right quadrant will have more samples than expected and PRKAB1 high will be associated with gene A high. For agonist, top-left and bottom-right quadrants are expected to be sparse. Therefore, $T_{index}$ for agonist is computed as follows:

$$T_{index} = \frac{1}{4}\left(\frac{0.3}{(S_{01}+1)} + \frac{0.3}{(S_{10}+1)} + p_{01} + p_{10}\right) \quad (12)$$

Therapeutic indices range from 0.36 to 0.027 where the most effective drug targets will be close to 0.027 and abandoned drug targets will be close to 0.36. Since all the currently known FDA-approved drug targets have therapeutic indices <0.1, we set this number as a threshold to identify effective drug targets. Lower therapeutic indices mean a stronger BIR with PRKAB1 which predicted phase III successes for many drugs in IBD[9]. Only four out of 16 targets have therapeutic indices <0.1. For effectiveness, we also check EMT and Inflammation scores in addition to the therapeutic index. Effective targets are observed to have better scores for both EMT and Inflammation, and they are likely to be present in both EMT and Inflammation Boolean paths. See the section "Identification of Epithelial-Mesenchymal and Inflammation-Fibrosis continuum".

*Target report card – IBD outcome.* Several datasets with annotations of IBD (normal vs IBD, GSE73661, GSE16879, GSE59071, GSE48958) as well as the aggressiveness of IBD such as active from inactive disease (GSE59071, GSE48958), responders from non-responders receiving two different biologics, Infliximab or Vedolizumab (GSE73661, GSE16879, GSE50594, E-MTAB-7604), and even distinguished those with the quiescent disease with or without remote neoplasia (GSE37283) were used to assess the strength of association of drug targets with IBD outcome. See the section "Generation of heat maps and drug targets score" of how drug targets to score is computed, samples are ordered and association with disease outcome is measured. A list of barplots is used to visualize the sample ordering and the association with disease outcome.

*Target report card – Network-prioritized mouse model.* Drug targets score is computed for the mouse model IBD dataset (DSS bulk GSE42768, DSS epithelium E-MTAB-5249, TNBS GSE53835, Citrobacter GSE90577, adoptive T-cell transfer ACT GSE87317, ACT GSE27302, IL10 -/- GSE39859, TNFR1 -/- GSE107933, TNFR2 -/- GSE65408) to test how combined gene expression values of the drug targets are associated with disease annotation. See the section "Generation of heat maps and drug targets score" of how drug targets score is computed, samples are ordered and association with disease outcome is measured. A list of barplots is used to visualize the sample ordering and the association with disease outcome.

*Target report card – estimation of gender bias.* A box plot of the gene expression values of the individual target gene is computed in the Peters-2017 GSE83687 dataset to test if there are significant gender-associated differences. The box plots of individual genes for both males and females are plotted side-by-side to visualize the differences.

*Target report card – Predicted tissue cell type of action.* It is important to know which cell types are relevant for the optimal action of drug targets. We assume that the drug action is dictated by cell type-specific expression patterns of the drug targets. We predict cell type-specific expression patterns using various techniques including correlation, standard deviation, and previously published MiDReG algorithm[93]. We assembled several gene expression databases for this task. A large human colon tissue database ($n = 1911$) was assembled by pooling several normal colon, adenoma, and colorectal cancer datasets from NCBI GEO (Supplementary Fig. 3a). All the samples in this database were analyzed using bulk tissue in Affymetrix U133 Plus 2.0 microarray platform. A large human colorectal cancer cell line database ($n = 264$) was prepared to identify genes expressed in epithelium because these are likely homogeneous and devoid of stromal tissue such as fibroblasts and immune cells. Microarray datasets of FACS purified macrophages, FACS purified GI fibroblasts, and FACS purified lymphocytes were downloaded using GSE134312, GSE63626, and GSE24759, respectively (Supplementary Fig. 3a). The algorithm that predicts whether a gene is expressed in top, bottom, lymphocytes, macrophages, and fibroblasts is described in a flow chart (Supplementary Fig. 3b). MiDReG algorithm is performed on the human colon tissue database ($n = 1911$) to predict top/bottom of the crypt marker[89,93]. Boolean implication "KRT20 low => X low" is used to predict the expression of gene X at the top of crypt[89]. Boolean implication "KRT20 low => X high" is used to predict the expression of gene X at the bottom of the crypt. Since the human colon tissue database (n=1911) contains bulk tissue samples, the expression of gene X is restricted to the epithelium by filtering the gene expression in the human colorectal cancer cell line database ($n = 264$). LGR5 correlation > 0.8 is performed in the FACS purified colon crypt dataset (GSE31255) to predict the bottom of the crypt markers independently. Standard deviation > 0.5 in human B cells and T cells (GSE24759) is used to predict lymphocyte-specific expression. Genes expressed in human macrophages and fibroblast are predicted by computing the StepMiner threshold on the bulk datasets GSE134312 and GSE63626, respectively, which is compared to the StepMiner threshold obtained in the original human global tissue dataset (GSE119087, $n = 25,955$). While both PPARG and PPARA are predicted to be expressed in top of the crypt and macrophages, PPARG is predicted to be expressed in fibroblasts in addition (Supplementary Fig. 3c).

*Construction of a Network of Boolean Implications.* A Boolean implication network (BIN) is created by identifying all significant pairwise BIRs that are consistent in both Peters-2017 GSE83687 and Arijs-2018 GSE73661 datasets independently (Supplementary Fig. 1a)[95,96]. The BIN contains the six possible Boolean relationships between genes in the form of a directed graph with nodes as genes and edges as the Boolean relationship between the genes. The nodes in the BIN are genes and the edges correspond to BIRs. Equivalent and Opposite relationships are denoted by undirected edges and the other four types (low => low; high => low; low => high; high => high) of BIRs are denoted by having a directed edge between them. The network of equivalences seems to follow a scale-free trend; however, other as we generated PAR5359 through two intermediate steps, symmetric relations in the network do not follow scale-free properties. BIR is strong and robust when the sample sizes are usually more than 200 (from our experience of using Boolean Implication for more than 10 years). All our previous papers use thousands of diverse samples to establish BIRs. Boolean Implication analysis is carried out for the first time in such a low number of samples such as the selected IBD GSE83687 dataset ($n = 134$). We have demonstrated that we have a reasonable False Discovery Rate (<0.001) when S > 2.5 and $p < 0.1$ are used. The IBD dataset was prepared for Boolean analysis by filtering genes that had a reasonable dynamic range of expression values. When the dynamic range of expression values was small, it was difficult to distinguish if the values were all low or all high or there were some high and some low values. Thus, it was determined to be best to ignore them during Boolean analysis. The filtering step was performed by analyzing the fraction of high and low values identified by the StepMiner algorithm[94]. Any probe set or genes which contained <5% of high or low values were dropped from the analysis.

*Generation of clustered Boolean implication network.* Clustering was performed in the BIN to dramatically reduce the complexity of the network (Supplementary Fig. 1b). A clustered Boolean implication network (CBIN) was created by clustering nodes in the original BIN by following the equivalent BIRs. One approach is to build connected components in a undirected graph of Boolean equivalences. However, because of noise, the connected components become internally inconsistent, e.g., two genes opposite to each other becomes part of the same connected component. In addition, the size of clusters became unusually big with almost everything in one cluster. To avoid such a situation, we need to break the component by removing the weak links. To identify the weakest links, we first computed a minimum spanning tree for the graph and computed Jaccard similarity coefficient for every edge in this tree. Ideally, if two members are part of the same

cluster they should share as many connections as possible. If they share less than half of their total individual connections (Jaccard similarity coefficient < 0.5) the edges are dropped from further analysis. Thus, many weak equivalences were dropped using the above algorithm leaving the clusters internally consistent. We removed all edges that have Jaccard similarity coefficient <0.5 and built the connected components with the rest. The connected components were used to cluster the BIN which is converted to the nodes of the CBIN. The distribution of cluster sizes was plotted in a log-log scale to observe the characteristic of the Boolean network. The clusters sizes were distributed along a straight line in a log-log plot suggesting scale-free properties. The choice of the threshold on the Jaccard similarity coefficient plays an important role in determining the size and the number of clusters as well as whether they are internally consistent. We found that a threshold of 0.5 gave us reasonable number of clusters and followed a scale-free distribution in the cluster sizes. A bigger threshold such as 0.7 to 0.9 will be very aggressive and reduce the cluster sizes (almost all edges will be dropped). A smaller number such as 0.4 will tend to make bigger cluster with unusual distribution of cluster sizes. A new graph was built that connected the individual clusters to each other using Boolean relationships. The link between two clusters (A, B) was established by using the top representative node from A that was connected to most of the members of A and sampling 6 nodes from cluster B and identifying the overwhelming majority of BIRs between the nodes from each cluster.

A CBIN was created using the selected Peters-2017 GSE83687 and Arijs-2018 GSE73661 datasets. Each cluster was associated with healthy or disease samples based on where these gene clusters were highly expressed. The edges between the clusters represented the Boolean relationships that are color-coded as follows: orange for low => high, dark blue for low => low, green for high => high, red for high => low, light blue for equivalent, and black for the opposite.

*Generation of IBD, UC, and CD maps.* IBD map is derived from the CBIN of the Peters-2017 GSE83687 and Arijs-2018 GSE73661 datasets by focusing on the largest clusters and their connections. A subset of the CBIN (Supplementary Fig. 1b) is constructed by following the top 10 largest clusters and a Boolean path sequence of for high => high, high => low, and low => low (dark blue). Machine learning is performed on this network to identify the Boolean path that can distinguish normal vs IBD samples. A Boolean path is converted to a path score as mentioned above using a linear combination of normalized gene expression values. The strength of classification of healthy and IBD samples using this score is computed by the ROC-AUC measurement. We performed a multivariate regression to identify the best Boolean path that predicts normal vs IBD samples in the cohort GSE6731 (4 N, 5 UC, 7 CD). Path #1-2-3 emerged as the winner. UC map (Supplementary Fig. 4) and CD map (Supplementary Fig. 5) are created by restricting the Peters-2017 GSE83687 dataset to UC only and CD only samples before constructing the CBIN respectively. Arijs-2018 GSE73661 dataset is not used for the UC and the CD maps.

*Generation of heat maps and drug targets score.* A composite score is computed as follows when many genes are considered drug targets which include a summary of their gene expression values. To compute the composite score, gene expression values were normalized according to a modified Z-score approach centered around StepMiner threshold (formula = (expr - SThr)/3*stddev). The samples were ordered according to an average of the normalized gene expression values in the given gene list. The heatmap uses red colors for the high values, white colors for the intermediate values, and blue colors for low values. Gene names for a few selected genes are highlighted on the left to show their expression patterns. Drug targets score is computed as a linear combination of the normalized gene expression values (the modified Z-score). Samples are ordered using the drug targets score and the strength of the association between gene expression and disease annotation is computed using ROC-AUC measurement. A barplot is used to visualize the sample ordering with different color codes for the disease annotation. In addition, a set of violin plots is used just below the barplot to demonstrate the distribution of the drug target score across different disease annotations.

*Identification of epithelial-mesenchymal and inflammation-fibrosis continuum.* Top genes involved with epithelial-mesenchymal processes and inflammation-fibrosis processed are chosen from the literature review, and used earlier[9]. Given a list of genes *BoNE* computes a subgraph of the CBIN graph by identifying clusters that include one or more genes from this list. BoNE then search for a path in this subgraph as mentioned before with the original CBIN graph. The path identified is used to draw a model of the gene expression timeline. The continuum is identified by computing a score based on the path.

*GeneSet Enrichment Analysis (GSEA).* GSEA was performed using python gseapy 0.10.2 package. The difference in average expression values of the two groups is used to compute the gene rank file. GSEA pre-ranked analysis is performed on the precomputed rank file to check the significance of the geneset enrichment score and generate the enrichment plot. GSEA computes four key statistics for the gene set enrichment analysis report: enrichment score (ES), normalized enrichment score (NES), false discovery rate (FDR), nominal *P* value.

*Measurement of classification strength or prediction accuracy.* Receiver operating characteristic (ROC) curves were computed by simulating a score based on the

ordering of samples that illustrates the diagnostic ability of binary classifier system as its discrimination threshold is varied along the sample order. The ROC curves were created by plotting the true positive rate (TPR) against the false positive rate (FPR) at various threshold settings. The area under the curve (often referred to as simply the AUC) is equal to the probability that a classifier will rank a randomly chosen IBD samples higher than a randomly chosen healthy samples. In addition to ROC AUC, other classification metrics such as accuracy ((TP + TN)/N; TP: True Positive; TN: True Negative; N: Total Number), precision (TP/(TP + FP); FP: False Positive), recall (TP/(TP+FN); FN: False Negative) and f1 (2 * (precision * recall)/(precision + recall)) scores were computed. Precision score represents how many selected items are relevant and recall score represents how many relevant items are selected. Fisher exact test is used to examine the significance of the association (contingency) between two different classification systems (one of them can be ground truth as a reference).

*Statistical analyses.* All statistical tests were performed using R version 3.2.3 (2015-12-10). Standard t-tests were performed using python scipy.stats.ttest_ind package (version 0.19.0) with Welch's Two Sample *t*-test (unpaired, unequal variance (equal_var=False), and unequal sample size) parameters. Multiple hypothesis corrections were performed by adjusting *p* values with statsmodels.stats.multitest.multiple tests (fdr_bh: Benjamini/Hochberg principles). The results were independently validated with R statistical software (R version 3.6.1; 2019-07-05). Pathway analysis of gene lists was carried out via the Reactome database and algorithm[97]. Reactome identifies signaling and metabolic molecules and organizes their relations into biological pathways and processes. Kaplan–Meier analysis is performed using lifelines python package version 0.22.8. Violin, Swarm, and Bubble plots are created using python seaborn package version 0.10.1.

## Experimental

*Reagents.* All reagents were purchased from Sigma-Aldrich (St. Louis, MO), unless otherwise indicated. Goat anti-rabbit and goat anti-mouse Alexa Fluor 680 and IRDye 800 F(ab')2 were purchased from LI-COR Biosciences (Lincoln, NE). Pioglitazone was purchased from Selleck Chemicals (Houston, TX). GW7647, GW6471, and GW9662 were purchased from Tocris Biosciences (Bristol, UK). PAR5359 was synthesized by the Yang's lab, Department of Chemistry and Biochemistry, University of California San Diego.

*Synthesis of PAR5359.* ethyl (S)-2-ethoxy-3-(4-(2-hydroxyethoxy) phenyl)propanoate (**compound 1**, Supplementary Fig. 6a): Ethylene carbonate (663 mg, 5.04 mmol, 3 equiv.) was added to the solution of ethyl (S)-2-ethoxy-3-(4 hydroxyphenyl) propanoate (400 mg, 1.68 mmol) and potassium carbonate ($K_2CO_3$) (695 mg, 5.04 mmol, 3 equiv.) in dry dimethylformamide (DMF) (5 mL). The reaction was stirred at 80 °C overnight (16 h). The reaction flask was diluted with ~ 50 mL of ethyl acetate (EtOAc), and the solids were removed with filtration through celite. Water (~ 30 mL) was added, and the solution mixture was extracted twice with EtOAc (~ 50 mL x 2), the combined organics were washed with brine (~ 50 mL), dried over $MgSO_4$, filtered, and concentrated *in vacuo*. The product was purified via $SiO_2$ column chromatography (using a gradient of 20 to 30% to 50% EtOAc in hexanes as eluent) to give the title compound **1** as a clear oil (335 mg, 70% yield). 1H NMR (300 MHz, $CDCl_3$) δ (ppm) = 7.16 (d, 2H), 6.84 (d, 2H), 4.16 (q, 2H), 4.04 (t, 2H), 3.98-3.91 (m, 3H), 3.64-3.54 (m, 1H), 3.38-3.28 (m, 1H), 2.95 (d, 2H), 2.23 (s, 1H), 1.21 (t, 3H), 1.15 (t, 3H).

*ethyl (S)-2-ethoxy-3-(4-(2-((methylsulfonyl)oxy)ethoxy)phenyl)propanoate* (**compound 2**, Supplementary Fig. 6b): Methanesulfonyl chloride (MsCl, 237 mg, 0.16 mL, 2.02 mmol, 1.7 equiv.) was added dropwise to an ice-cold solution of compound **1** (335 mg, 1.19 mmol) and triethylamine (TEA) (240 mg, 0.331 mL, 2.38 mmol, 2 equiv.) in dry dichloromethane (DCM) (7 mL). The reaction was stirred at room temperature for 2.5 h, then diluted with ~50 mL of 1 M HCl aq. solution. The aqueous layer was then extracted with DCM (50 mL x 2), the combined organic layers were washed with sequence of ~50 mL of saturated $NaHCO_3$ aq. solution, ~50 mL of water, and ~ 50 mL of brine. The organic layer was dried over $MgSO_4$, filtered, and concentrated in vacuo, to give the title compound **2** as a brown oil, and no further purification (412 mg, 96% yield). 1H NMR (300 MHz, $CDCl_3$) δ ppm = 7.19 (d, 2H), 6.83 (d, 2H), 4.57 (t, 2H), 4.22 (t, 2H), 4.16 (t, 2H), 3.97 (q, 1H), 3.66-3.55 (m, 1H) 3.40-3.30 (m, 1H), 3.09 (s, 3H), 2.97 (d, 2H), 1.24 (t, 3H), 1.16 (t, 3H).

*ethyl (S)-3-(4-(2-(4-(4-chlorophenyl)-3,6-dihydropyridin-1(2H)-yl)ethoxy) phenyl)-2-ethoxypropanoate* (**compound 3**, Supplementary Fig. 6c): 4-(4-chlorophenyl)-1,2,3,6tetrahydropyridine (314 mg, 1.37 mmol, 1.2 equiv.), sodium iodide (NaI) (34 mg, 0.23 mmol, 0.2 equiv.), and potassium carbonate ($K_2CO_3$) (471 mg, 3.42 mmol, 3 equiv.) was added to the solution of compound **2** (412 mg, 1.14 mmol) in dry DMF (6 mL). The reaction was stirred at 60 °C overnight (16 h). The reaction flask was then diluted with ~50 mL of EtOAc, and the solids were removed with filtration through celite. Water (~30 mL) was added, and the solution mixture was extracted three times with EtOAc (~50 mL x 3), the combined organics were washed with brine (~50 mL), dried over $MgSO_4$, filtered, and concentrated *in vacuo*. The product was purified via $SiO_2$ column chromatography (using a gradient of 20 to 30% to 40% EtOAc in hexanes as eluent) to give the title compound **3** as a clear oil (189 mg, **36% yield**). 1H NMR (300 MHz, $CDCl_3$) δ ppm = 7.33 (d, 2H), 7.28 (d, 2H), 7.17 (d, 2H), 6.86 (d, 2H), 6.06 (s, br, 1H), 4.20-

4.08 (m, 3H), 3.97 (t, 1H), 3.65-3.55 (m, 1H), 3.40-3.32 (m, 1H), 3.31 (d, 2H), 2.94 (t, 4H), 2.86 (t, 2H), 2.57 (s, br, 2H), 1.23 (t, 3H), 1.16 (t, 3H).

(S)-3-(4-(2-(4-(4-chlorophenyl)-3,6-dihydropyridin-1(2H)-yl)ethoxy)phenyl)-2-ethoxypropanoic acid (**PAR5359**, Supplementary Fig. 6d): Lithium hydroxide monohydrate (26 mg, 0.624 mmol, 2 equiv.) was added to the solution of compound **3** (163 mg, 0.312 mmol) in tetrahydrofuran (THF) (6 mL) and water (1.5 mL). The reaction was stirred at room temperature for 4 h and was then quenched by addition of ~1 mL 1 M HCl aq. solution. The reaction flask was then evaporated to dryness *in vacuo*. The resultant solids were purified via $SiO_2$ column chromatography (using a gradient of 4 to 6% to 10% MeOH in DCM as eluent) to give the title compound **PAR5359** as a white solid (109 mg, 71% yield) 1H NMR (500 MHz, CD3OD-d4) δ ppm = 7.25-7.23 (d, 2H), 7.13-7.11 (d, 2H), 6.98-6.97 (d, 2H), 6.70-6.68 (d, 2H), 5.94 (t, J = 2.2 Hz, 1H), 4.15-4.13 (m, 2H), 3.74-3.73 (m, 2H), 3.60-3.59 (m, 2H), 3.38-3.32 (m, 5H), 3.05-2.98 (m, 1H), 2.74-2.71 (m, 1H), 2.63-2.55 (m, 3H), 0.84 (t, J= 7.0 Hz, 3H); 13 C NMR (126 MHz, CD3OD-d4) δ ppm = 181.1, 158.7, 139.6, 136.4, 135.7, 134.2, 132.5, 132.4, 130.6, 130.5, 128.7, 128.6, 119.0, 118.9, 116.2, 116.2, 84.6, 67.4, 64.7, 57.1, 53.3, 51.7, 40.6, 26.5, 16.3; ESI-MS: 430.2 $[M+H]^+$.

*Bacteria and bacterial culture*. For bacterial culture adherent Invasive *Escherichia coli* strain LF82 (AIEC-LF82), *Citrobacter rodentium* (strain DBS100), and *Salmonella enterica serovar typhimurium* (*strain SL1344*), a single colony was inoculated into LB broth and grown for 6-8 h on shaking incubator, followed by overnight culture under oxygen-limiting conditions, but without shaking, to maintain their pathogenicity as done previously[98–100]. Cells were infected with bacteria with indicated MOI in figure legends.

*C. rodentium induced infectious colitis and in vivo treatments*. *C. rodentium* (strain DBS100) induced infectious colitis studies were performed on 7-week-old C57BL/6 mice. Mice were obtained from Jackson Laboratories (Bay Harbor, ME) and housed in animal facility for 1 week to acclimatize before using for the experiment. All animals were housed and euthanized according to the University of California San Diego Institutional Animal Care and Use Committee (IACUC) policies and guidelines. *C. rodentium* were grown overnight in LB broth with shaking at 37 °C and mice were gavaged orally with $5 \times 10^8$ CFU in 0.1 ml of PBS[51,53]. To determine viable bacterial numbers in faeces, fecal pellets were collected from individual mice, homogenized in cold PBS, serially diluted, and plated on MacConkey agar plates. The number of CFU was determined after overnight incubation at 37 °C. Colon samples were collected to assess histology and levels of mRNA (by qPCR). For treatment study, PPARα agonist (GW7647, 20 μg/kg body weight/day), PPARγ agonist (Pioglitazone, 20 μg/kg body weight/day), PPARα/γ dual agonist (PAR5359, 1 mg/kg body weight/day) were administered via intraperitoneal route in 200 μl total volume (DMSO <4%).

*DSS-induced colitis*. For DSS-colitis experiments, 7-week-old C57BL/6 mice were obtained from Jackson Laboratories (Bay Harbor, ME). All animals were housed and euthanized according to the University's IACUC policies and guidelines. Colitis was induced by oral administration of 2.5% dextran sulfate sodium (DSS, w/v) (MP Biomedicals, MW 36–50 kDa) in drinking water for five days[101,102]. PAR5359 (1 mg/kg/day) was administered via intrarectal route in a total volume of 50 μl dissolved in DMSO (the final concentration of DMSO was <4%). Mice were hung upside-down for 30 s post-injection to ensure that the injected solution was retained in the colon. Mice were sacrificed on the 9th day, and colon length was assessed. Colon samples were collected for assessing gene expression (by qPCR). Water levels were monitored to determine the volume of water consumed by all groups. Each animal was monitored for animal weight loss, stool consistency, and fecal blood and these parameters were used to calculate an average Disease Activity Index (DAI)[103,104]. Colon histology was assessed in samples stained with hematoxylin using standard protocols. Histological scores were scored in a blinded manner using H&E stained colonic tissue[105]. Briefly, histological score is a cumulative score of intestinal inflammatory cell infiltrate (mild-1, moderate-2, and marked-3) and epithelial architecture (Focal erosion-1, Focal erosion+ulcerations-2, and extended ulceration and granulations-3).

*Thioglycolate-elicited murine peritoneal macrophages generations*. Thioglycolate-elicited murine peritoneal macrophages (TGPMs) were isolated from 8- to 12-week-old C57BL/6 mice and cultured[66]. All animals were housed and euthanized according to the University of California San Diego IACUC policies and guidelines. Briefly, cells were collected from peritoneal lavage with ice-cold RPMI (10 ml per mouse) 4 days after intraperitoneal injection of 3 ml of aged, sterile 3% thioglycolate broth (BD Difco, USA). Cells were filtered with 70 μ filter, centrifuged, and resuspended in RPMI-1640 containing 10% FBS and 1% penicillin/streptomycin. TGPMs were plated with the required cell density and the media was changed after 4 h to remove non adherent cells. Depending on the experiment, TGPMs were seeded in 6-well ($2 \times 10^6$ cells/well), 12-well ($1 \times 10^6$ cells/well), or 24-well plates ($5 \times 10^5$ cells/well) with appropriate and consistent cell densities. TGPMs were allowed to adjust to overnight culture before the addition of stimuli: LPS (10–100 ng/ml) in the presence or absence of PPAR agonists and antagonists as described in figure legends.

*Measurement of ROS*. To assess whether PPAR agonists modulate bacteria (LF82 and SL)-induced ROS in peritoneal macrophages were (50,0000 cells/96 well) were treated with *AIEC*-LF82/SL in presence or absence of PPAR agonists/inhibitors. The redox-sensitive, cell-permeable dihydroethidium (hydroethidine or DHE) was used to detect the cellular production of ROS (ROS, Detection Cell-Based Assay Kit, Cayman Chemical) and the plate was read using fluorescence microplate reader (ex 500–530 nm/em 590–620 nm).

*Gentamicin protection assay*. Quantification of viable intracellular bacteria was done by using the gentamicin protection assay[98]. Briefly, Peritoneal macrophage TGPMs, $2 \times 10^5$ cells per well were seeded into 24-well culture dishes overnight before infection at an MOI of 10 for 1 h in antibiotic-free RPMI media containing 10% FBS in a 37 °C $CO_2$ incubator. Cells were then washed and incubated with gentamicin (200 μg/ml) for 90 min to kill extracellular bacteria. Further, cells were washed and incubated with antibiotic-free RPMI media containing 10% FBS for 1-24 h and subsequently lysed in 1% Triton-X 100, lysates were serially diluted and plated on LB agar plates. Total CFU were counted after overnight incubation at 37 °C. To test effect of PPAR agonists, cells were pre-treated for 30 min with PPAR agonists.

*RNA extraction and real-time quantitative PCR (RT-qPCR)*. Total RNA was isolated using TRIzol reagent (Life Technologies) and Quick-RNA MiniPrep Kit (Zymo Research, USA). The amount and purity of RNA were measured using NanoDrop® spectrophotometer (Thermo Fisher, USA). RNA was converted into cDNA using the qScript™ cDNA SuperMix (Quantabio). Quantitative RT-PCR (qPCR) was carried out in 384-well plate using PowerUp™ SYBR™ green master mix (Applied Biosystems, USA) and performed on the SteepOnePlus Quantitative platform (Life Technologies, USA). The cycle threshold (Ct) of target genes was normalized to 18S rRNA gene and the fold change in the mRNA expression was determined using the $2^{-\Delta\Delta Ct}$ method using either LPS or bacteria treated samples for the normalization in the respective experiment. Primers used in RT-qPCR reactions were those that have been previously validated in similar studies[106–109] and primer sequences are stated in the supplementary key resource table (Supplementary Table 6).

*RNA-seq library preparation*. Sequencing libraries were generated using the Illumina TruSeq Stranded Total RNA Library Prep Gold with TruSeq Unique Dual Indexes (Illumina, San Diego, CA)[110]. Libraries were amplified and sequenced on an Illumina NovaSeq 6000 by the Institute of Genomic Medicine (IGM) at the University of California San Diego.

*Cytokine assays*. Cytokines including TNFa, IL6, IL1b, and IL10 were measured in cell supernatant using ELISA MAX Deluxe kits from Biolegend.

**Statistics and reproducibility**. Experimental values are presented as the means of replicate experiments ±SEM. Statistical analyses were performed using GraphPad Prism software version 8.0 (GraphPad Software). Differences between the two groups were evaluated using Student's *t*-test (parametric) or Mann–Whitney *U*-test (non-parametric). To compare more than three groups, one-way analysis of variance followed by Tukey's post hoc test was used. Differences at $P < 0.05$ were considered significant. Please see Supporting Information for details regarding Boolean data analysis.

**Study approval**
*Human subjects*. Blood samples were obtained from either healthy volunteers or from IBD patients undergoing colonoscopies a part of their routine care and follow-up at UC San Diego's Inflammatory Bowel Disease (IBD) Center. Patients were recruited and consented using a study proposal approved by the Institutional Review Board of the University of California, San Diego. Isolation of blood monocytes was carried out using an approved human research protocol (IRB# 160246) that covers human subject research at the UC San Diego HUMANOID Center of Research Excellence (CoRE). The clinical phenotype and information were curated based on histopathology reports from Clinical Pathology and Chart check, followed by consultation with a specialist at UC San Diego's IBD Center.

*Animals*. All animal studies were approved by the University of California, San Diego Institutional Animal Care and Use Committee (IACUC). Adult C57BL/6 mice were acquired from Jackson Laboratories. All animals were maintained in institutional animal care. Provided with standard light–dark cycle, fed with standard laboratory chow and clean drinking water.

**Reporting summary**. Further information on research design is available in the Nature Research Reporting Summary linked to this article.

## Data availability
The data supporting the findings of this study are available within the paper is available in Supplementary Data 1. RNA sequencing data that support the findings of this study have been deposited in NCBI GEO with the accession number GSE171057. Publicly available datasets used: GSE83687, GSE73661, GSE16879, GSE59071, GSE48958,

GSE50594, GSE37283, E-MTAB-7604, GSE42768, E-MTAB-5249, GSE53835, GSE90577, GSE87317, GSE27302, GSE39859, GSE107933, GSE65408, GSE73661, GSE16879, GSE59071, GSE48958, GSE59071, GSE48958, GSE73661, GSE16879, GSE50594, E-MTAB-7604, GSE37283, GSE134312, GSE63626, GSE24759, GSE31255, GSE24759, GSE134312, GSE63626, GSE119087.

## Code availability

The codes are publicly available at the following links: https://github.com/sahoo00/BoNE; https://github.com/sahoo00/Hegemon.

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

## Acknowledgements

We thank Dharanidhar Dang (UCSD) for comments and critiques during the preparation of the manuscript. This work was supported by National Institutes of Health (NIH) grants R01-AI141630 (to PG), DK107585 (to SD). PG, SD, and DS were also supported by the Leona M. and Harry B. Helmsley Charitable Trust and the NIH (UG3TR003355, UG3TR002968, and R01-AI55696). GDK was supported through The American Association of Immunologists Intersect Fellowship Program for Computational Scientists and Immunologists. J.S. acknowledges support from the Interfaces Training Grant at UCSD (NIH T32EB009380). Authors thank Lee Swanson, Courtney Tindle, Stella-Rita Ibeawuchi, Julian Tam, and Madhubanti Mullick for their comments, feedback, and technical support. This manuscript includes data generated at the UC San Diego Institute of Genomic Medicine (IGC) using an Illumina NovaSeq 6000 that was purchased with funding from a National Institutes of Health SIG grant (#S10 OD026929). Additionally, a P30 grant (NIH/NIDDK, P30DK120515) subsidized the RNA Seq and histology work showcased here. Figures 1, 2, 3f, 4a, 5a, 6a, 7, and 8 were created with BioRender.com.

## Author contributions

P.G., D.S., S.D., and G.D.K. conceptualized, supervised, administered the project, and acquired funding to support it. G.D.K., I.M.S., M.S.A., E.V., F.U., and J.R.S. were involved in data curation and formal analysis. D.S. developed computational modeling and software for analysis and performed all computational analysis. G.D.K., I.M.S., M.S.A., E.V., and F.U. conducted animal studies for colitis. G.D.K., I.M.S., M.S.A., E.V., F.U., and S.D. performed cell and tissue analysis including qPCR, ROS, bacterial clearance, ELISA. D.T. assisted G.D.K. for qPCR and analysis. J.R.S., G.L., and J.Y. prepared PAR5359 compound. W.J.S. provided key resources for human subjects. D.S., G.D.K., and P.G. prepared figures for data visualization, wrote the original draft, reviewed and edited the manuscript. All co-authors approved the final version of the manuscript.

## Competing interests

The authors declare the following competing interests: S.D., D.S., and P.G. have a patent on the methodology. Barring this, the authors have declared that no conflict of interest exists.
