## [Peer Review File · Communications Biology]

Reviewers' comments:

Reviewer #1 (Remarks to the Author):

Brief summary of the manuscript

The Authors using Boolean Implication Network (BoNE), a toolkit to explore and analyze the directed graph representation of biological datasets, built a predictive map for Inflammatory Bowel Disease (IBD) to construct a performing work-flow to develop a therapeutic strategy that involves dual agonism of two nuclear receptors, PPAR α / γ . This in silico analysis had allowed them to construct a solid rationale for set-up experiments to demonstrate that the simultaneous and balanced agonistic activation of the PPAR α / γ as an effective barrier protective strategy in IBD and that macrophages are one of the primary target cell types of this therapeutic strategy.

Overall impression of the work

The manuscript is an original opening in my opinion new vision of drug discovery for IBD and in general for overall chronic disease. The manuscript is well written and organized. The Authors with very clear and summarizing images render the paper readable and captivating. Anyway, some questions should be addressed.

Specific comments, with recommendations for addressing each comment

In the section on methods, there is missing information. Regarding, *C. rodentium* induced infectious colitis it is not clear which kind of mice the Authors used and the age and how the Authors performs oral administration, by drinking water, by gavage?. Also regarding the other models, it is not clear if the drug delivery was done intrarectal alone or in combination with intraperitoneal. In addition the sentence "All animals were housed and euthanized according to University of California San Diego Institutional Animal Care and Use Committee (IACUC) policies and guidelines" should be added also for the other model! In the paragraph of TGPMs, we have to be more precious what means required cell density? As well as the LPS doses 10-100ng/ml which does the author choose to perform the experiments.

Finally regarding PCR I suggest writing RT-qPCR. The Authors omitted how they checked the RNA quality. The second point regarding PCR is the Y-axis, in figure 5 D the Authors put Relative expression (normalized to 18S and not $2^{-\Delta\Delta Ct}$) there is not a calibrator that usually has 1 as value; in fig6 the Authors put Relative gene expression ($2^{-\Delta Ct}$) that is not Livak methods and also in this plot there is not the calibrator. I'm a bit puzzled by the Real-time PCR methods and how the results have been extrapolated. Moreover, it is necessary to add in supplementary Materials the PCR efficiency.

Finally in Fig4E the image of H&E-stained colon tissues. Mag = 200x for PPAR α / γ looks smaller. Regarding statistics, there are not the number of biological replicates. Please add the scale bar.

Reviewer #2 (Remarks to the Author):

Brief Summary

The authors utilize an AI algorithm taking in various publicly available transcriptional data sets in IBD as well as collecting samples from patients to identify PPAR α and PPAR γ to be therapeutic targets for IBD. They identified the animal models and cell types to perturb the PPAR signaling using a synthesized dual agonist. They adopted various animal models and in vitro models to study the effects of the agonists. In general, the work is rigorous and of high quality. The manuscript should be of broad interest because it combines computational algorithm for therapeutic target identifications, animal models/patient samples, and macrophage biology. There are various concerns listed below that need to be clarified. Most importantly, the authors claim that PPAR α is pro-inflammatory while the literature suggests otherwise. Repeating the experiment in Fig. 6 with additional controls and functional assessments should confirm the contrary nature of the results.

Major:

1. The authors suggest that PPAR α activation is pro-inflammatory. Multiple groups in various models have shown that activation of PPAR α has anti-inflammatory effects (PMCID: PMC7140404, PMC2095111, PMID: 29547450).
2. Can the authors comment on the AI results/ data/ and prior literature regarding Tight junction?

In Supp Fig 8, PPAR α agonist induces no changes from DMSO yet it is predicted in the AI algorithm Supp Fig 1 to play a role. Conversely, PPAR γ agonist increased tight junction while not predicted in Supp Fig 1 and they have shown barrier protective effects in UC patients (Line 171).

3. Supp Fig 3C, there are no cell type specific expression patterns for PPAR γ and PPAR α is stated in the legend.

4. Figure 4 and Supp7B Any more histology scores other than images of H&E stain? It will strengthen the argument with quantitative scoring (statistics) and of individual categories of histopathological parameters.

5. Figure 4: Since Macrophage M1 v M2 cluster was included in the figure, can authors comment on what the analysis suggests? The RNAseq analysis was performed from colon. What is the relative population of macrophages (since it is the major cell type being focused on)? And if the authors use their algorithm to look at macrophage-specific expression from bulk samples as they performed previously, can they identify more macrophage-specific clusters that are dysregulated in the *C. rodentium* colitis model?

6. Supp Fig 9, what are the histology scoring based on? 9 days DSS-induced colitis is a bit short after 7 days DSS induction.

7. Figure 5: in line 373, authors stated that "PAR5359 did not interfere with the production of microbe-induced ROS...". Based on Figure 5C, it is significantly decreased from DMSO control. Again, showing whether the dual agonist is significantly different from individual agonists would be helpful (since multiple comparisons were performed).

8. The authors stated that "pioglitazone appeared to suppress IL10 mRNA, it did not suppress the levels, suggesting that Ppar- γ agonist is sufficient for an overall anti-inflammatory phenotype." It is difficult to draw conclusion since pioglitazone did not induce any significant changes in either RNA or protein levels (Fig. 5D/E).

9. Unsure the accuracy of the title that PPAR γ delays bacterial clearance (Supp Fig 10) since individual PPAR agonists had no significant effects on CFU counts. Only the dual agonist decreased CFU. As for the cytokine levels, only PPAR γ and dual agonist significant changed the levels. Regarding statistics, since One-way ANOV, multiple comparisons were performed, it would be informative to show whether the effects of the dual agonist is significantly different from the individual agonists. Lastly, in Supp Fig 10, it is a little strange that in IL6, PPAR α agonist decreased the levels more drastically from DMSO than PPAR γ or dual agonists and the error bar does not seem larger than dual agonist and yet it was not significant.

10. Line 394-5 "Ppar- α . agonism enhances the induction of inflammation and ROS, and promotes bacterial clearance...". I think this claim is overstated. While it's true that PPAR α activation decreased CFU and increased ROS (marginally) in the AIEC-LF82 infection (Fig. 5), it did not affect CFU in Salmonella infection (Supp Fig 10) and it had not significant changes in any of the transcriptional expression or protein production (not significant; other than IL10) (Fig. 5/Supp Fig 10).

11. Similar experiments using macrophages and PPAR inhibitors have been published with different results (PMCID: PMC4467528, PMID: 25881202). Generally, LPS in combination with PPAR α agonist decreased inflammatory response and LPS + PPAR α agonist and antagonist alleviate the decrease. I think the experiments in Fig. 6 should be repeated with this modification since it is contrary to other previously published data. PPAR activations (qRT-PCR of target genes, protein analysis of PPAR[levels and modifications phosphorylation/SUMOylating] should also be assessed to ensure that the chemical perturbations are altering PPAR signaling as intended.

Minor:

1. Abstract Line 60: Remove comma after (PAR5359) in

2. Line 118: Add Figure numbers (1) before Step 3/4

3. Line 155/162 and throughout the manuscript: Since these are proteins, nomenclature conventions would be PPAR α and PPAR γ not Ppar- α and Ppar- γ

4. Supp Fig 7A "Colon excised See Figure 3". Should be Figure 4

5. Line 559, 560: Strain names for *C. rodentium* and *Salmonella Typhimurium* should be stated

REVIEWERS' COMMENTS:

Reviewer #1 (Remarks to the Author):

Brief summary of the manuscript

The Authors using Boolean Implication Network (BoNE), a toolkit to explore and analyze the directed graph representation of biological datasets, built a predictive map for Inflammatory Bowel Disease (IBD) to construct a performing work-flow to develop a therapeutic strategy that involves dual agonism of two nuclear receptors, PPAR α / γ . This in silico analysis had allowed them to construct a solid rationale for set-up experiments to demonstrate that the simultaneous and balanced agonistic activation of the PPAR α / γ as an effective barrier protective strategy in IBD and that macrophages are one of the primary target cell types of this therapeutic strategy.

Overall impression of the work

The manuscript is an original opening in my opinion new vision of drug discovery for IBD and in general for overall chronic disease. The manuscript is well written and organized. The Authors with very clear and summarizing images render the paper readable and captivating. Anyway, some questions should be addressed.

Specific comments, with recommendations for addressing each comment

In the section on methods, there is missing information. Regarding, *C. rodentium* induced infectious colitis it is not clear which kind of mice the Authors used and the age and how the Authors performs oral administration, by drinking water, by gavage?. Also regarding the other models, it is not clear if the drug delivery was done intrarectal alone or in combination with intraperitoneal. In addition the sentence "All animals were housed and euthanized according to University of California San Diego Institutional Animal Care and Use Committee (IACUC) policies and guidelines" should be added also for the other model! In the paragraph of TGPMs, we have to be more precious what means required cell density? As well as the LPS doses 10-100ng/ml which does the author choose to perform the experiments.

Finally regarding PCR I suggest writing RT-qPCR. The Authors omitted how they checked the RNA quality. The second point regarding PCR is the Y-axis, in figure 5 D the Authors put Relative expression (normalized to 18S and not $2^{-\Delta\Delta Ct}$) there is not a calibrator that usually has 1 as value; in fig6 the Authors put Relative gene expression ($2^{-\Delta Ct}$) that is not Livak methods and also in this plot there is not the calibrator. I'm a bit puzzled by the Real-time PCR methods and how the results have been extrapolated. Moreover, it is necessary to add in supplementary Materials the PCR efficiency.

Finally in Fig4E the image of H&E-stained colon tissues. Mag = 200x for PPAR α / γ looks smaller. Regarding statistics, there are not the number of biological replicates. Please add the scale bar.

Reviewer #2 (Remarks to the Author):

Brief Summary

The authors utilize an AI algorithm taking in various publicly available transcriptional data sets in IBD as well as collecting samples from patients to identify PPAR α and PPAR γ to be therapeutic targets for IBD. They identified the animal models and cell types to perturb the PPAR signaling using a synthesized dual agonist. They adopted various animal models and in vitro models to study the effects of the agonists. In general, the work is rigorous and of high quality. The manuscript should be of broad interest because it combines

computational algorithm for therapeutic target identifications, animal models/patient samples, and macrophage biology. There are various concerns listed below that need to be clarified. Most importantly, the authors claim that PPAR α is pro-inflammatory while the literature suggests otherwise. Repeating the experiment in Fig. 6 with additional controls and functional assessments should confirm the contrary nature of the results.

Major:

1. The authors suggest that PPAR α activation is pro-inflammatory. Multiple groups in various models have shown that activation of PPAR α has anti-inflammatory effects (PMCID: PMC7140404, PMC2095111, PMID: 29547450).
2. Can the authors comment on the AI results/ data/ and prior literature regarding Tight junction? In Supp Fig 8, PPAR α agonist induces no changes from DMSO yet it is predicted in the AI algorithm Supp Fig 1 to play a role. Conversely, PPAR γ agonist increased tight junction while not predicted in Supp Fig 1 and they have shown barrier protective effects in UC patients (Line 171).
3. Supp Fig 3C, there are no cell type specific expression patterns for PPARG and PPARG is stated in the legend.
4. Figure 4 and Supp7B Any more histology scores other than images of H&E stain? It will strengthen the argument with quantitative scoring (statistics) and of individual categories of histopathological parameters.
5. Figure 4: Since Macrophage M1 v M2 cluster was included in the figure, can authors comment on what the analysis suggests? The RNAseq analysis was performed from colon. What is the relative population of macrophages (since it is the major cell type being focused on)? And if the authors use their algorithm to look at macrophage-specific expression from bulk samples as they performed previously, can they identify more macrophage-specific clusters that are dysregulated in the *C. rodentium* colitis model?
6. Supp Fig 9, what are the histology scoring based on? 9 days DSS-induced colitis is a bit short after 7 days DSS induction.
7. Figure 5: in line 373, authors stated that "PAR5359 did not interfere with the production of microbe-induced ROS...". Based on Figure 5C, it is significantly decreased from DMSO control. Again, showing whether the dual agonist is significantly different from individual agonists would be helpful (since multiple comparisons were performed).
8. The authors stated that "pioglitazone appeared to suppress Il10 mRNA, it did not suppress the levels, suggesting that Ppar- γ agonist is sufficient for an overall anti-inflammatory phenotype." It is difficult to draw conclusion since pioglitazone did not induce any significant changes in either RNA or protein levels (Fig. 5D/E).
9. Unsure the accuracy of the title that PPAR γ delays bacterial clearance (Supp Fig 10) since individual PPAR agonists had no significant effects on CFU counts. Only the dual agonist decreased CFU. As for the cytokine levels, only PPAR γ and dual agonist significant changed the levels. Regarding statistics, since One-way ANOV, multiple comparisons were performed, it would be informative to show whether the effects of the dual agonist is significantly different from the individual agonists. Lastly, in Supp Fig 10, it is a little strange that in IL6, PPAR α agonist decreased the levels more drastically from DMSO than PPAR γ or dual agonists and the error bar does not seem larger than dual agonist and yet it was not significant.
10. Line 394-5 "Ppar- α . agonism enhances the induction of inflammation and ROS, and promotes bacterial clearance...". I think this claim is overstated. While it's true that PPAR α activation decreased CFU and increased ROS (marginally) in the AIEC-LF82 infection (Fig. 5), it did not affect CFU in Salmonella infection (Supp Fig 10) and it had not significant changes in any of the transcriptional expression or protein production (not significant; other than IL10) (Fig. 5/Supp Fig 10).
11. Similar experiments using macrophages and PPAR inhibitors have been published with different results (PMCID: PMC4467528, PMID: 25881202). Generally, LPS in combination with PPAR α agonist decreased inflammatory response and LPS + PPAR α agonist and antagonist alleviate the decrease. I think the experiments in Fig. 6 should be repeated with this modification since it is contrary to other previously published data. PPAR activations (qRT-PCR of target genes, protein analysis of PPAR[levels and modifications phosphorylation/SUMOylating] should also be assessed to ensure that the chemical perturbations are altering

PPAR signaling as intended.

Minor:

1. Abstract Line 60: Remove comma after (PAR5359) in
2. Line 118: Add Figure numbers (1) before Step 3/4
3. Line 155/162 and throughout the manuscript: Since these are proteins, nomenclature conventions would be PPAR α and PPAR γ not Ppar- α and Ppar- γ
4. Supp Fig 7A "Colon excised See Figure 3". Should be Figure 4
5. Line 559, 560: Strain names for *C. rodentium* and *Salmonella Typhimurium* should be stated

POINT BY POINT RESPONSE TO REVIEWER'S COMMENTS

(Original comments are in **black**; responses are in **blue**)

Reviewers' comments:

We thank both the reviewers for providing positive comments to improve the quality of manuscript.

Reviewer #1 (Remarks to the Author):

Brief summary of the manuscript

The Authors using Boolean Implication Network (BoNE), a toolkit to explore and analyze the directed graph representation of biological datasets, built a predictive map for Inflammatory Bowel Disease (IBD) to construct a performing work-flow to develop a therapeutic strategy that involves dual agonism of two nuclear receptors, PPAR α / γ . This in silico analysis had allowed them to construct a solid rationale for set-up experiments to demonstrate that the simultaneous and balanced agonistic activation of the PPAR α / γ as an effective barrier protective strategy in IBD and that macrophages are one of the primary target cell types of this therapeutic strategy.

RESPONSE: We thank the Reviewer #1 for the thoughtful review of our work.

Overall impression of the work

The manuscript is an original opening in my opinion new vision of drug discovery for IBD and in general for overall chronic disease. The manuscript is well written and organized. The Authors with very clear and summarizing images render the paper readable and captivating. Anyway, some questions should be addressed.

RESPONSE: We appreciate the encouraging comments and the generous praise.

Specific comments, with recommendations for addressing each comment

In the section on methods, there is missing information. Regarding, *C. rodentium* induced infectious colitis it is not clear which kind of mice the Authors used and the age and how the Authors performs oral administration, by drinking water, by gavage?. Also regarding the other models, it is not clear if the drug delivery was done intrarectal alone or in combination with intraperitoneal.

RESPONSE: We agree that these details should have been there. We apologize for this unintentional omission.

Action(s) taken: We have included the recommended details. For the convenience of the Editor/Reviewer, we copy-pasted the sentence below:

566 ***C. rodentium* induced infectious colitis and *in vivo* treatments**

567 *C. rodentium* induced infectious colitis studies were performed on 7-week old C57BL/6 mice. Mice were
568 obtained from Jackson Laboratories (Bay Harbor, ME) and housed in animal facility for 1-week to acclimatize
569 before using for the experiment. All animals were housed and euthanized according to the University of
570 California San Diego Institutional Animal Care and Use Committee (IACUC) policies and guidelines. *C.*
571 *rodentium* were grown overnight in LB broth with shaking at 37 °C and mice were gavaged orally with 5 x
572 10⁸ CFU in 0.1 ml of PBS as described (51, 53). To determine viable bacterial numbers in faeces, fecal pellets

In addition the sentence “All animals were housed and euthanized according to University of California San Diego Institutional Animal Care and Use Committee (IACUC) policies and guidelines” should be added also for the other model!

Action(s) taken: Agree; we have now done that.

582 For DSS-colitis experiments, 7-week old C57BL/6 mice were obtained from Jackson Laboratories (Bay
583 Harbor, ME). All animals were housed and euthanized according to the University’s IACUC policies and
584 guidelines. Colitis was induced by oral administration of 2.5% dextran sulfate sodium (DSS, w/v) (MP
585 Biomedicals, MW 36–50 kDa) in drinking water for five days as described (86, 87). PAR5359 (1 mg/kg/day)

598 **Thioglycolate-elicited murine peritoneal macrophages generations**

599 Thioglycolate-elicited murine peritoneal macrophages (TGPMs) were isolated from 8- to 12-week-old
600 C57BL/6 mice and cultured as described previously(66). All animals were housed and euthanized according
601 to the University of California San Diego IACUC policies and guidelines. Briefly, cells were collected from
602 peritoneal lavage with ice cold RPMI (10 ml per mouse) 4 days after intraperitoneal injection of 3 ml of aged,

In the paragraph of TGPMs, we have to be more precious what means required cell density? As well as the LPS doses 10-100ng/ml which does the author choose to perform the experiments.

Action(s) taken: We have now added these details in the manuscript.

Finally regarding PCR I suggest writing RT-qPCR. The Authors omitted how they checked the RNA quality.

Action(s) taken: We agree. In this revised version of the manuscript, we have added all these missing details. For the convenience of the Editor/Reviewer, we copy-pasted the sentence below:

628 **RNA extraction and real time quantitative PCR (RT-qPCR)**

629 Total RNA was isolated using TRIzol reagent (Life Technologies) and Quick-RNA MiniPrep Kit (Zymo
630 Research, USA). Amount and purity of RNA was measured using NanoDrop® spectrophotometer (Thermo
631 Fisher, USA). RNA was converted into cDNA using the qScript™ cDNA SuperMix (Quantabio).

The second point regarding PCR is the Y-axis, in figure 5 D the Authors put Relative expression (normalized to 18S and not $2^{-\Delta\Delta Ct}$) there is not a calibrator that usually has 1 as value; in fig6 the Authors put Relative gene expression ($2^{-\Delta Ct}$) that is not Livak methods and also in this plot there is not the calibrator. I'm a bit puzzled by the Real-time PCR methods and how the results have been extrapolated. Moreover, it is necessary to add in supplementary Materials the PCR efficiency.

RESPONSE: We apologize for the confusion in our presentation. We thank reviewer for pointing different Y-axis label in Fig 5D and Fig6 for RT-qPCR analysis.

Action(s) taken: To avoid confusion, we corrected the Y-axis label in Fig 5D and Fig6 for RT-qPCR analysis and clarified in the legends and method section. For the convenience of the Editor/Reviewer, we copy-pasted the sentence below. As for primer PCR efficiency, we want to emphasize that for standard targets such as these, it is customary that we use primers that have been previously published and validated (PMID: 33055214, PMID: 16150853). We have now inserted this information in the text.

633 (Applied Biosystems, USA) and performed on the SteepOnePlus Quantitative platform (Life Technologies,
634 USA). The cycle threshold (Ct) of target genes was normalized to 18S rRNA gene and the fold change in the
635 mRNA expression was determined using the $2^{-\Delta\Delta Ct}$ method using either LPS or bacteria treated samples
636 for the normalization in the respective experiment. Primers used in RT-qPCR reactions were those that have
637 been previously validated in similar studies (91-94) and primer sequences are stated in the supplementary key
638 resource table.

Finally, in Fig4E the image of H&E-stained colon tissues. Mag = 200x for PPAR α / γ looks smaller. Regarding statistics, there are not the number of biological replicates. Please add the scale bar.

RESPONSE: We thank reviewer for pointing out these errors. Leica1000 LED microscope attached with ICC50HD camera is used to capture the images using 10x and 20x objectives. Appropriate changes are made in the manuscript to avoid confusion.

Action(s) taken: Figure legend (Figure 4) is modified to: 1) include the number of biological replicates; 2) explain details of statistics and 3) scale bar is added to the images and defined in the legend. For the convenience of the Editor/Reviewer, we copy-pasted the sentence below.

1027 the indicated readouts. **(B-D)** Line graphs in B display time series of the burden of viable bacteria in feces. Scatter plots with bar
1028 graphs in C compare the peak burden of viable bacteria in feces on day 7. Scatter plots with bar graphs in D display the area under
1029 the curve (AUC) for the line graph in B. **(E)** Images display representative fields from H&E-stained colon tissues of 4-5 mice in
1030 each group. Mag = 100x (top) and 200x (bottom); Scale bars, 100 μm . White arrowheads point to immune cell infiltrates. **(F)**
1031 Stacked bar graphs display four parameters of inflammation that were quantified in H&E stained colon tissues in E. Statistics: All
1032 results are displayed as mean \pm SEM. Significance was tested on cumulative histological score using one-way ANOVA followed
1033 by Tukey's test for multiple comparisons. Significance: ns, non-significant, p-value less than 0.05 was considered significant. **(G)**
1034 Violin plots (left) display the deviation of expression of genes in Clusters #1-2-3 in the IBD network, as determined by RNA Seq
1035 on murine colons. Bar plot (right) displays the rank ordering of the samples. Welch's t-test was used to determine statistical
1036 significance. Significance: ns, non-significant, p-value less than 0.05 was considered significant. **(H)** Pre-ranked GSEA based on
1037 pairwise differential expression analyses (DMSO vs PAR5359 groups) are displayed as enrichment plots for epithelial tight (top)
1038 and adherens (middle) junction signatures and balanced macrophage processes (bottom). See also **Supplemental Figure 7** for the
1039 Day #7 results in the *C. rodentium*-induced colitis model, **Supplemental Figure 8** for extended GSEA analyses, and **Supplemental**
1040 **Figure 9** for the effect of PAR5359 on DSS-induced colitis in mice.

Reviewer #2 (Remarks to the Author):

Brief Summary

The authors utilize an AI algorithm taking in various publicly available transcriptional data sets in IBD as well as collecting samples from patients to identify PPAR α and PPAR γ to be therapeutic targets for IBD. They identified the animal models and cell types to perturb the PPAR signaling using a synthesized dual agonist. They adopted various animal models and in vitro models to study the effects of the agonists. In general, the work is rigorous and of high quality. **The manuscript should be of broad interest because it combines computational algorithm for therapeutic target identifications, animal models/patient samples, and macrophage biology.**

RESPONSE: We thank reviewer for appreciating our work and for the encouraging remarks.

There are various concerns listed below that need to be clarified. Most importantly, the authors claim that PPAR α is pro-inflammatory while the literature suggests otherwise. Repeating the experiment in Fig. 6 with additional controls and functional assessments should confirm the contrary nature of the results.

Major:

1. The authors suggest that PPAR α activation is pro-inflammatory. Multiple groups in various models have shown that activation of PPAR α has anti-inflammatory effects (PMCID: PMC2095111, PMID: 29547450).

RESPONSE: We apologize for not clarifying this ambiguous point. In the original submitted version of this manuscript, we explicitly stated that – “*Noteworthy, while the role of PPAR γ in colitis has been investigated through numerous studies over the past 3 decades (11-13) (Supplemental Table 1), the role of PPAR α has been contradictory (Supplemental Table 2), ...*”. In Supplementary Table 2 we had summarized findings of several studies on PPAR α , which were split on their conclusion whether PPAR α is pro- vs anti-inflammatory in the setting of colitis. However, it is possible that the supplementary nature of such inclusion ultimately makes it harder for readers to appreciate the existing literature. We had missed these two citations suggested by the reviewer because they are focused on lung infection with non-enteric pathogens.

Action taken: To ensure that important information and citations are accessible in the main text, we have now included key citations in the main text. For the convenience of the reviewer, we have copied-pasted the section below.

162 Noteworthy, while the role of PPAR γ in colitis has been investigated through numerous studies over
163 the past 3 decades (11-13) (**Supplemental Table 1**), the role of PPAR α has been contradictory (**Supplemental**
164 **Table 2**), and their dual agonism in IBD has never been explored. **The studies on PPAR α are equally split on**
165 **whether it is pro- or anti-inflammatory in action (14-20). By contrast, all studies on PPAR γ agree that its**
166 **agonism ameliorates DSS-induced colitis (13, 21, 22).** Although claimed to be effective on diverse cell types
167 in the gut (epithelium, T-cells, and macrophages), the most notable target cells of PPAR γ agonists are

2. Can the authors comment on the AI results/ data/ and prior literature regarding Tight junction? In Supp Fig 8, PPAR α agonist induces no changes from DMSO yet it is predicted in the AI algorithm Supp Fig 1 to play a role. Conversely, PPAR γ agonist increased tight junction while not predicted in Supp Fig 1 and they have shown barrier protective effects in UC patients (Line 171).

RESPONSE: In this comment, we assume that the reviewer points to our own prior work (Sahoo D. et al., Nat Comm. 2021) when he/she says ‘AI results/data/prior literature’. If so, in that work, we had developed and validated a Boolean Map of IBD and used AI/ML algorithms to identify the gene clusters (upregulation of all three clusters, #1-2-3) that are indicative of healthy gut mucosal barrier function. In this work, that map of IBD is being leveraged. The same AI algorithm predicted that both PPAR α and PPAR γ should improve mucosal barrier function. Figure S1 simply shows the location of PPAR α /PPAR γ /PGC1 α within clusters 2 and 3, and the prominent reactome pathways enriched in those clusters are showcased (these are not predictions of the role of any particular gene at the TJs). Upregulation of all three genes [PPARA/PPARG/PGC1A] is predicted to improve mucosal barrier function/integrity. Our approach considers mucosal barrier as an entity that is comprised of the epithelial lining (and its TJs), the immune and non-immune cells in the lamina propria and the layer of mucin.

Thus, it appears that there is an unfortunate misunderstanding; we had never implied that one gene or the other was more (or less) likely to impact the TJs; all three were predicted to synergistically improve mucosal barrier functions. We believe that the reason why the dual PPAR α / γ agonist performs better in improving the mucosal barrier than the a/g single agonists is because PPARs act within different cell compartments in the mucosa, and their dual agonism in macrophages enhances bacterial clearance with controlled inflammation.

Action taken: We have now carefully gone over the Introduction and confirmed that we have indeed maintained “mucosal barrier” as the consistent terminology throughout when describing the functions of the IBD map derived gene clusters.

3. Supp Fig 3C, there are no cell type specific expression patterns for PPARG and PPARA is stated in the legend.

RESPONSE: We apologize for not clarifying for the readers which cell types may be the site of action of dual PPAR α / γ agonism.

Action(s) taken: In the revised version of the manuscript, we have taken 3 steps to ensure that this important information is not buried.

- 1) **Figure S3C:** We have revised this figure and explicitly stated the sites of action of PPAR α / γ next to the colon crypt.
- 2) We expanded the legend for Figure 3C to include the inference from this computational analysis.

3) We edited the main text to cite panels Figure S3A-C.

For the convenience of the reviewer, we have copied and pasted the **Fig S3C** and legend below.

Legend for 3C: (C) Cell type specific expression patterns for PPARG and PPARA is computed using the flow chart in panel B. Their actions potentially overlap in the epithelium at the crypt top and in macrophages, and hence, any pharmacologic impact of their dual agonism may be appreciated in these cell types.

4. Figure 4 and Supp7B Any more histology scores other than images of H&E stain? It will strengthen the argument with quantitative scoring (statistics) and of individual categories of histopathological parameters.

RESPONSE: We agree.

Action taken: In this revised version of the manuscript, we have additionally carried out quantitative assessment (with statistics) for histological scores in our *Citrobacter colitis* model. The findings are included as a

new **Figure panel 4F**. More specifically, we assessed 4 parameters (bulleted below). Statistical analyses showed that the dual PPAR agonist PAR5359 was effective in reducing all the 4 parameters of inflammation that we assessed quantitatively.

- Submucosal inflammation
- Percent area involved
- Inflammatory infiltrates in LP
- Crypt hyperplasia

For the convenience of the reviewer, we have copied and pasted the newly added figure panels in **Fig 4F** and the figure legend.

103
103
103
103

5. Figure 4: Since Macrophage M1 v M2 cluster was included in the figure, can authors comment on what the analysis suggests? The RNAseq analysis was performed from colon. What is the relative population of macrophages (since it is the major cell type being focused on)? And if the

authors use their algorithm to look at macrophage-specific expression from bulk samples as they performed previously, can they identify more macrophage-specific clusters that are dysregulated in the *C. rodentium* colitis model?

RESPONSE: This comment is about the GSEA analysis in Figure 4G in the original submission (which is Figure 4H in the revised submission). We appreciate this thoughtful 2-part question.

In the **first part**, the reviewer wonders what the GSEA analysis suggests and ask us to elaborate on the M1 vs M2 macrophage polarization states in the RNA seq dataset. We apologize that we had not done that in the original submission. What this analysis represents is that samples are compared for up-regulated genes distinguishing between M1 (pro-inflammatory) and M2 (anti-inflammatory) macrophage subtypes. We found by GSEA analysis that these genes were not differentially upregulated in the PAR5359-treated samples vs the DMSO treated controls (Fig 4H, bottom).

In the **second part**, he/she wonders if we use computational approaches that we used in Figure 2F/3B-C (to pinpoint cell type specific action of PPAR α /g) can help identify macrophage-specific gene clusters that are dysregulated in *C. rodentium* model. We appreciate this insightful question. While the approaches we developed can estimate macrophage abundance, they are not geared to assess macrophage processes in bulk seq samples as accurately as would be possible using single cell Seq.

Actions taken: Part 1 of the comment is addressed by expanding the results section. For the convenience of the reviewer, we have copied and pasted the newly added sentence in the “Results” section.

319 agonists Pioglitazone or GW7647 was able to significantly preserve epithelial junction signatures (both tight
320 and adherens junctions) while maintaining balanced macrophage processes (compare **Figure 4H** with
321 **Supplemental Figure 8A**). **These enrichment analyses confirmed that, compared to DMSO-treated infected**
322 **colons, the epithelial junction related genes were significantly enriched in dual PPAR α /g-agonist treated**
323 **samples (Figure 4H, top and middle) without significant changes in macrophage polarization (Figure 4H,**
324 **bottom).** These findings are in keeping with the predictions that epithelial cells and macrophages maybe the

6. Supp Fig 9, what are the histology scoring based on? 9 days DSS-induced colitis is a bit short after 7 days DSS induction.

RESPONSE: This is a 2-part question.

In the **first part** the reviewer asks what was the basis for the histology scoring. thank reviewer for pointing this important scoring details. To clarify, histology score that we presented in the original version of the manuscript for DSS colitis was a cumulative score of intestinal inflammatory cell infiltrates (mild-1, moderate-2 and marked-3) and epithelial architecture (Focal erosion-1, Focal erosion+ulcerations-2 and extended ulceration and granulations-3). These are standardized methods published earlier [Erben U, Lodenkemper C, Doerfel K, Spieckermann S, Haller D, Heimesaat MM, et al. A guide to histomorphological evaluation of intestinal inflammation in mouse models. *Int J Clin Exp Pathol.* 2014;7(8):4557-76.]

In the **second part**, the reviewer raises a concern that sac on the 9th day after a 7-day DSS induction may be too short. In this regard, we would like to point out that we simply followed the acute DSS colitis protocol that is well established in the field. In this model, induction is typically 5-7 days and the animals are sacrificed on day 9 (PMID: 28569761).

Action(s) taken: Details of histological score is mentioned in methods under the section of- '**DSS-induced colitis**'. For reviewer's convenience, we copy-pasted the edited text from revised manuscript.

591 consistency, and fecal blood and these parameters were used to calculate an average Disease Activity Index
592 (DAI) as described previously (88, 89). Colon histology was assessed in samples stained with hematoxylin
593 using standard protocols. Histological scores were scored in a blinded manner using H&E stained colonic
594 tissue as described previously (90). Briefly, histological score is a cumulative score of intestinal inflammatory
595 cell infiltrate (mild-1, moderate-2 and marked-3) and epithelial architecture (Focal erosion-1, Focal
596 erosion+ulcerations-2 and extended ulceration and granulations-3).

7. Figure 5: in line 373, authors stated that “PAR5359 did not interfere with the production of microbe-induced ROS...”. Based on Figure 5C, it is significantly decreased from DMSO control. Again, showing whether the dual agonist is significantly different from individual agonists would be helpful (since multiple comparisons were performed).

RESPONSE: The reviewer is right. Based on reviewer’s suggestions, we have now edited panel Fig 5C to show the statistical significance of all conditions compared to other groups. We also edited the main text to make it more clear for the reader. These new statistical comparisons have allowed us to modify our claim and re-state the findings more accurately.

Actions taken: For reviewer’s convenience we have copied-pasted the section of the manuscript where these edits appear. We believe that these descriptions now more accurately reflect the results.

372 component for effective bacterial killing (68, 69). By contrast, GW7647 did not reduce ROS induction,
373 whereas PAR5359 treatment permitted an intermediate amount of induction of ROS (significantly lower than
374 GW7647; Figure 5C). Thus, the dual agonist PAR5359 allowed effective bacterial clearance (like GW7647;
375 Figure 5B) with minimal ROS induction (like Pioglitazone; Figure 5C).

8. The authors stated that “pioglitazone appeared to suppress IL10 mRNA, it did not suppress the levels, suggesting that Ppar- γ agonist is sufficient for an overall anti-inflammatory phenotype.” It is difficult to draw conclusion since pioglitazone did not induce any significant changes in either RNA or protein levels (Fig. 5D/E).

RESPONSE: We apologize for this ambiguous statement; we agree that, as stated, it is neither meaningful, nor an accurate reflection of the results. What we found, is that neither IL10 mRNA, nor IL10 protein was suppressed by the dual agonist.

Action(s) taken: In the revised manuscript, we have simplified our description of results to call out on the major finding, i.e., dual agonist suppresses proinflammatory cytokines, both mRNA and protein, without suppressing anti-inflammatory IL10 cytokine. For reviewer’s convenience we have copied-pasted the section of the manuscript where these edits appear.

383 also translated to the abundance of secreted cytokines released by the macrophages in the supernatant. Thus,
384 both mRNA and protein estimations agreed that PPAR α / γ dual agonism inhibits pro-inflammatory *Il1 β* and
385 *Il6* cytokines while permitting the induction of the major anti-inflammatory cytokine *Il10*. Similar findings
386 were also observed in the case of another enteric pathogen, Salmonella (*S. enterica*); it is noteworthy that
387 unlike AIEC-LF82, *S. enteritica* is more efficient in surviving and multiplying inside the macrophage. The
388 dual PPAR5359 agonist enhanced bacterial clearance (Supplemental Figure 10A-B) with limited induction of
389 pro-inflammatory cytokines (significantly lower than control) and full augmentation of anti-inflammatory *Il10*
390 cytokine (similar to control) (Supplemental Figure 10C). Neither PPAR α -, nor PPAR γ - single agonists
391 achieved this desirable profile, i.e., effective bacterial clearance with controlled inflammation. PPAR α -agonist
392 improved clearance, but failed to reduce proinflammatory cytokines. By contrast, PPAR γ -agonist failed to
393 promote clearance, but significantly reduced all cytokines (Supplemental Figure 10).

394 Taken together, these studies on enteric pathogens show that- (i) PPAR γ agonism induces ‘tolerance’
395 by suppressing inflammation, inhibiting ROS production and delaying bacterial clearance; (ii) PPAR α
396 agonism induces ‘reactivity’ by promoting bacterial clearance, permitting the full extent of ROS production
397 and the induction of proinflammatory cytokines, but suppressing the anti-inflammatory *il10* cytokine; and (iii)
398 PPAR α / γ dual agonism achieves a more balanced response; it suppresses proinflammatory cytokines without
399 suppressing anti-inflammatory cytokine *Il10*, and thereby, permits the extent of inflammation and ROS
400 induction that is optimal and sufficient to promote bacterial clearance.

9. Unsure the accuracy of the title that PPAR γ delays bacterial clearance (Supp Fig 10) since individual PPAR agonists had no significant effects on CFU counts. Only the dual agonist decreased CFU. As for the cytokine levels, only PPAR γ and dual agonist significant changed the levels. Regarding statistics, since One-way ANOV, multiple comparisons were performed, it would be informative to show whether the effects of the dual agonist is significantly different from the individual agonists. Lastly, in Supp Fig 10, it is a little strange that in IL6, PPAR α agonist decreased the levels more drastically from DMSO than PPAR γ or dual agonists and the error bar does not seem larger than dual agonist and yet it was not significant.

RESPONSE: We thank reviewer for the careful evaluation and valuable suggestions to improve the accuracy and clarity of our manuscript. There are multiple parts of this comment, all pertaining to Supplementary Figure 10.

- The reviewer questions the accuracy of the title that PPAR γ delays bacterial clearance (Supp Fig 10) since individual PPAR agonists had no significant effects on CFU counts. Only the dual agonist decreased CFU.
 - o **Response:** We agree.
 - o **Action(s) taken:** In the revised manuscript we edited the supp figure 10 title

141 **Supplementary Figure 10: PPAR α / γ -dual agonists enhance bacterial (*Salmonella enteritica*) clearance.** (A) Schematic displays
142 the experimental design and workflow. Thioglycolate-induced murine peritoneal macrophages (TG-PM) pretreated with PPAR agonists
143 (see box, below; 20 nM GW7647, 10 μ M Pioglitazone and 1 μ M PAR5359) were infected with *Salmonella enterica* (MOI 10) and
144 subsequently analyzed at 6 h post-infection for bacterial count (Gentamicin protection assay) and secretion of inflammatory cytokines
145 (in supernatant media by ELISA). (B) Bar graphs show percent internalized viable bacterial counts of *Salmonella enterica*. (C) Bar
146 graphs display the extent of secreted cytokines (IL1 β , IL6, TNF α and IL10) in the supernatant media. Statistics: One-way ANOVA

- The reviewer points out that as for the cytokine levels, only PPAR γ and dual agonist significant changed the levels. Regarding statistics, since One-way ANOV, multiple comparisons were performed, it would be informative to show whether the effects of the dual agonist is significantly different from the individual agonists. Lastly, in Supp Fig 10, it is a little strange that in IL6, PPAR α agonist decreased the levels more drastically from DMSO than PPAR γ or dual agonists and the error bar does not seem larger than dual agonist and yet it was not significant.
 - o **Response:** We apologize for the confusion caused due to the absence of multi-way comparisons/statistics. We agree that such comparisons would help.
 - o **Action(s) taken:** In the revised manuscript, we have included statistical comparisons for all conditions that were relevant in assessing the data. We also simplified the way the results are described. For reviewer's convenience we have copied-pasted the section of the manuscript where these edits appear.

383 also translated to the abundance of secreted cytokines released by the macrophages in the supernatant. **Thus,**
384 **both mRNA and protein estimations agreed that PPAR α / γ dual agonism inhibits pro-inflammatory *Il1 β* and**
385 ***Il6* cytokines while permitting the induction of the major anti-inflammatory cytokine *Il10*. Similar findings**
386 **were also observed in the case of another enteric pathogen, *Salmonella* (*S. enterica*); it is noteworthy that**
387 **unlike *AIEC-LF82*, *S. enteritica* is more efficient in surviving and multiplying inside the macrophage. The**
388 **dual PAR5359 agonist enhanced bacterial clearance (Supplemental Figure 10A-B) with limited induction of**
389 **pro-inflammatory cytokines (significantly lower than control) and full augmentation of anti-inflammatory *Il10***
390 **cytokine (similar to control) (Supplemental Figure 10C). Neither PPAR α -, nor PPAR γ - single agonists**
391 **achieved this desirable profile, i.e., effective bacterial clearance with controlled inflammation. PPAR α -agonist**
392 **improved clearance, but failed to reduce proinflammatory cytokines. By contrast, PPAR γ -agonist failed to**
393 **promote clearance, but significantly reduced all cytokines (Supplemental Figure 10).**

394 Taken together, these studies on enteric pathogens show that- (i) PPAR γ agonism induces 'tolerance'
395 by suppressing inflammation, inhibiting ROS production and delaying bacterial clearance; (ii) PPAR α
396 agonism induces 'reactivity' by promoting bacterial clearance, permitting the full extent of ROS production
397 and the induction of proinflammatory cytokines, but suppressing the anti-inflammatory *il10* cytokine; and (iii)
398 PPAR α / γ dual agonism achieves a more balanced response; it suppresses proinflammatory cytokines without
399 suppressing anti-inflammatory cytokine *Il10*, and thereby, permits the extent of inflammation and ROS
400 induction that is optimal and sufficient to promote bacterial clearance.

10. Line 394-5 “Ppar- α . agonism enhances the induction of inflammation and ROS, and promotes bacterial clearance...”. I think this claim is overstated. While it’s true that PPAR α activation decreased CFU and increased ROS (marginally) in the AIEC-LF82 infection (Fig. 5), it did not affect CFU in Salmonella infection (Supp Fig 10) and it had not significant changes in any of the transcriptional expression or protein production (not significant; other than IL10) (Fig. 5/Supp Fig 10).

RESPONSE: This comment is about a 2-part statement with 2 claims about PPAR α -agonists. We thank reviewer for asking us to revisit the data and perform statistical tests. We have now done that systematically for all figures and displayed the p values in each figure panel. These additional analyses showed that our second part of the initial claim is true, i.e., PPAR α -agonist indeed enhanced bacterial clearance compared to DMSO control. These additional analyses also made us recognize the need to change the first part of the claim, because, compared to DMSO conditions, PPAR α -agonist *did not* enhance the induction of ROS; nor did it enhance proinflammatory cytokines. In fact, there was no statistically significant differences between DMSO and PPAR α -agonist conditions for all assays assessing the proinflammatory cytokines and ROS.

Action(s) taken: In the revised manuscript, we have rephrased our statements. For reviewer’s convenience we have copied-pasted the section of the manuscript where these edits appear.

383 also translated to the abundance of secreted cytokines released by the macrophages in the supernatant. **Thus,**
384 **both mRNA and protein estimations agreed that PPAR α / γ dual agonism inhibits pro-inflammatory *III β* and**
385 ***II6* cytokines while permitting the induction of the major anti-inflammatory cytokine *III0*. Similar findings**
386 **were also observed in the case of another enteric pathogen, Salmonella (*S. enterica*); it is noteworthy that**
387 **unlike AIEC-LF82, *S. enteritica* is more efficient in surviving and multiplying inside the macrophage. The**
388 **dual PAR5359 agonist enhanced bacterial clearance (Supplemental Figure 10A-B) with limited induction of**
389 **pro-inflammatory cytokines (significantly lower than control) and full augmentation of anti-inflammatory *III0***
390 **cytokine (similar to control) (Supplemental Figure 10C). Neither PPAR α -, nor PPAR γ - single agonists**
391 **achieved this desirable profile, i.e., effective bacterial clearance with controlled inflammation. PPAR α -agonist**
392 **improved clearance, but failed to reduce proinflammatory cytokines. By contrast, PPAR γ -agonist failed to**
393 **promote clearance, but significantly reduced all cytokines (Supplemental Figure 10).**

394 Taken together, these studies on enteric pathogens show that- (i) PPAR γ agonism induces ‘tolerance’
395 by suppressing inflammation, inhibiting ROS production and delaying bacterial clearance; (ii) PPAR α
396 agonism induces ‘reactivity’ by promoting bacterial clearance, permitting the full extent of ROS production
397 and the induction of proinflammatory cytokines, but suppressing the anti-inflammatory *il10* cytokine; and (iii)
398 PPAR α / γ dual agonism achieves a more balanced response; it suppresses proinflammatory cytokines without
399 suppressing anti-inflammatory cytokine *III0*, and thereby, permits the extent of inflammation and ROS
400 induction that is optimal and sufficient to promote bacterial clearance.

11. Similar experiments using macrophages and PPAR inhibitors have been published with different results (PMCID: PMC4467528, PMID: 25881202). Generally, LPS in combination with PPAR α agonist decreased inflammatory response and LPS + PPAR α agonist and antagonist alleviate the decrease. I think the experiments in Fig. 6 should be repeated with this modification since it is contrary to other previously published data. PPAR activations (qRT-PCR of target genes, protein analysis of PPAR[levels and modifications

phosphorylation/SUMOylating] should also be assessed to ensure that the chemical perturbations are altering PPAR signaling as intended.

RESPONSE: We thank reviewer for suggesting references and suggestions to improve our manuscript.

We have now thoroughly looked up not just these two manuscripts, but also related work. Here is what we found. Both manuscripts use a PPAR α antagonist (GW6471) as a key reagent to manipulate the macrophages in their experiments. GW6471 strongly enhances the binding affinity of PPAR α ligand-binding domain to the co-repressor proteins SMRT and NCoR, and thereby, prevents PPAR α from assuming its active conformation (PMID: 11845213). This could explain why PPAR α inhibition inhibits inflammatory response by preventing transcriptional activity of PPAR α and without affecting PPAR levels and modifications phosphorylation/SUMOylating. In addition, PPAR α agonists used in these studies (fenofibrate and gemfibrozil) has been reported to have PPAR α independent mechanism(s).

We chose to study live microbes because it is the physiologically relevant stimuli in the gut lumen. It is expected to fully capture the innate immune sensing and signaling mechanisms beyond just the LPS/TLR4 axis. LPS and response to the same is likely to be elicited in our system. To confirm that the chemical perturbations we use are having the intended effect (i.e., transcriptional activation of the target PPAR), we conducted careful studies (see Supplementary Figure 6) to verify that prior to use in our work. We agree with the reviewer that studying phosphoregulation /SUMOylation of PPARs is likely to give us more insights; however, we believe that this is beyond the scope of this work (and not directly relevant to the claims herein).

Minor:

1. Abstract Line 60: Remove comma after (PAR5359) in

Action(s) taken: corrected in the revised manuscript.

2. Line 118: Add Figure numbers (1) before Step 3/4

Action(s) taken: corrected in the revised manuscript.

3. Line 155/162 and throughout the manuscript: Since these are proteins, nomenclature conventions would be PPAR α and PPAR γ not Ppar- α and Ppar- γ

Action(s) taken: corrected in the revised manuscript.

4. Supp Fig 7A “Colon excised See Figure 3”. Should be Figure 4

Action(s) taken: corrected in the revised Supp Fig 7A.

5. Line 559, 560: Strain names for C. rodentium and Salmonella Typhimurium should be stated

RESPONSE: We agree. This has been included in this revised manuscript.

REVIEWERS' COMMENTS:

Reviewer #1 (Remarks to the Author):

The manuscript is an original opening in my opinion new vision of drug discovery for IBD and in general for overall chronic disease. The manuscript is well written and organized. The Authors with very clear and summarizing images render the paper readable and captivatin

Reviewer #2 (Remarks to the Author):

Thank you for addressing this reviewer's concern and suggestions. Please add in the strain name for *C. rodentium* in line 568

ORIGINAL COMMENTS REVIEWERS

REVIEWERS' COMMENTS:

Reviewer #1 (Remarks to the Author):

The manuscript is an original opening in my opinion new vision of drug discovery for IBD and in general for overall chronic disease. The manuscript is well written and organized. The Authors with very clear and summarizing images render the paper readable and captivating.

Reviewer #2 (Remarks to the Author):

Thank you for addressing this reviewer's concern and suggestions. Please add in the strain name for *C. rodentium* in line 568

POINT BY POINT RESPONSE TO REVIEWER'S COMMENTS

(Original comments are in **black**; responses are in **blue**)

We are glad that the editors and referees accepted manuscript for the publications and are grateful for their thoughtful and constructive comments to improve the manuscript.

Reviewers' comments:

Reviewer #1 (Remarks to the Author):

The manuscript is an original opening in my opinion new vision of drug discovery for IBD and in general for overall chronic disease. The manuscript is well written and organized. The Authors with very clear and summarizing images render the paper readable and captivating.

RESPONSE: We appreciate the generous praise and help to improve our manuscript.

Reviewer #2 (Remarks to the Author):

Thank you for addressing this reviewer's concern and suggestions. Please add in the strain name for *C. rodentium* in line 568

RESPONSE: We thank reviewer and glad that reviewer accepted our manuscript. We added the strain name for *Citrobacter* in the revised manuscript. For the convenience of the Editor/Reviewer, we copy-pasted the sentence below:

924	Bacteria and bacterial culture
925	For bacterial culture adherent Invasive Escherichia coli strain LF82 (AIEC-LF82), Citrobacter rodentium
926	(strain DBS100) and Salmonella enterica serovar typhimurium (strain SL1344), a single colony was
927	inoculated into LB broth and grown for 6-8 h on shaking incubator, followed by overnight culture under